# Successor Uncertainties: Exploration and Uncertainty in Temporal Difference Learning

**David Janz**[*][†]
University of Cambridge
dj343@cam.ac.uk

**Jiri Hron**[*]
University of Cambridge
jh2084@cam.ac.uk

**Przemysław Mazur**
Wayve Technologies

**Katja Hofmann**
Microsoft Research

**José Miguel Hernández-Lobato**
University of Cambridge
Alan Turing Institute
Microsoft Research

**Sebastian Tschiatschek**
Microsoft Research

## Abstract

Posterior sampling for reinforcement learning (PSRL) is an effective method for balancing exploration and exploitation in reinforcement learning. Randomised value functions (RVF) can be viewed as a promising approach to scaling PSRL. However, we show that most contemporary algorithms combining RVF with neural network function approximation do not possess the properties which make PSRL effective, and provably fail in sparse reward problems. Moreover, we find that propagation of uncertainty, a property of PSRL previously thought important for exploration, does not preclude this failure. We use these insights to design Successor Uncertainties (SU), a cheap and easy to implement RVF algorithm that retains key properties of PSRL. SU is highly effective on hard tabular exploration benchmarks. Furthermore, on the Atari 2600 domain, it surpasses human performance on 38 of 49 games tested (achieving a median human normalised score of 2.09), and outperforms its closest RVF competitor, Bootstrapped DQN, on 36 of those.

## 1 Introduction

Perhaps the most important open question within reinforcement learning is how to effectively balance exploration of an unknown environment with exploitation of the already accumulated knowledge (Kaelbling et al., 1996; Sutton et al., 1998; Busoniu et al., 2010). In this paper, we study this in the classic setting where the unknown environment is modelled as a Markov Decision Process (MDP).

Specifically, we focus on developing an algorithm that combines effective exploration with neural network function approximation. Our approach is inspired by *Posterior Sampling for Reinforcement Learning* (PSRL; Strens, 2000; Osband et al., 2013). PSRL approaches the exploration/exploitation trade-off by explicitly accounting for uncertainty about the true underlying MDP. In tabular settings, PSRL achieves impressive results and close to optimal regret (Osband et al., 2013; Osband & Van Roy, 2016). However, many existing attempts to scale PSRL and combine it with neural network function approximation sacrifice the very aspects that make PSRL effective. In this work, we examine several of these algorithms in the context of PSRL and:

1. Prove that a previous avenue of research, propagation of uncertainty (O'Donoghue et al., 2018), is neither sufficient nor necessary for effective exploration under posterior sampling.

---

[*]Equal contribution
[†]Work partly done during an internship at Microsoft Research Cambridge

2. Introduce *Successor Uncertainties* (SU), a cheap and scalable model-free exploration algorithm that retains crucial elements of the PSRL algorithm.

3. Show that SU is highly effective on hard tabular exploration problems.

4. Present Atari 2600 results: SU outperforms Bootstrapped DQN (Osband et al., 2016a) on $36/49$ and Uncertainty Bellman Equation (O'Donoghue et al., 2018) on $43/49$ games.

## 2 Background

We use the following notation: for $X$ a random variable, we denote its distribution by $P_X$. Further, if $f$ is a measurable function, then $f(X)$ follows the distbution $f_\# P_X$ (the pushforward of $P_X$ by $f$).

We consider finite MDPs: a tuple $(\mathcal{S}, \mathcal{A}, \mathcal{T})$, where $\mathcal{S}$ is a finite state space, $\mathcal{A}$ a finite action space, and $\mathcal{T} \colon \mathcal{S} \times \mathcal{A} \to \mathcal{P}(\mathcal{S} \times \mathcal{R})$ a transition probability kernel mapping from the state-action space $\mathcal{S} \times \mathcal{A}$ to the set of probability distributions $\mathcal{P}(\mathcal{S} \times \mathcal{R})$ on the product space of states $\mathcal{S}$ and rewards $\mathcal{R} \subset \mathbb{R}$; $\mathcal{R}$ is assumed to be bounded throughout. For each time step $t \in \mathbb{N}$, the agent selects an action $A_t$ by sampling from a distribution specified by its policy $\pi \colon \mathcal{S} \to \mathcal{P}(\mathcal{A})$ for the current state $S_t$, and receives a new state and reward $(S_{t+1}, R_{t+1}) \sim \mathcal{T}(S_t, A_t)$. This gives rise to a Markov process $(S_t, A_t)_{t \geq 0}$ and a reward process $(R_t)_{t \geq 1}$. The task of solving an MDP amounts to finding a policy $\pi^\star$ which maximises the expected return $\mathbb{E}(\sum_{\tau=0}^\infty \gamma^\tau R_{\tau+1})$ with $\gamma \in [0, 1)$.

Crucial to many so called *model-free methods* for solving MDPs is the state-action value function (*Q function*) for a policy $\pi$: $Q_t^\pi := \mathbb{E}_t(\sum_{\tau=t}^\infty \gamma^{\tau-t} R_{\tau+1}) = \mathbb{E}_t(R_{t+1}) + \gamma \mathbb{E}_t(Q_{t+1}^\pi)$, where $\mathbb{E}_t$ is used to denote an expectation conditional on $(S_\tau, A_\tau)_{\tau \leq t}$. Model-free methods use the recursive nature of the Bellman equation to construct a model $\hat{Q}^\pi \colon \mathcal{S} \times \mathcal{A} \to \mathbb{R}$, which estimates $Q_t^\pi$ for any given $(S_t = s, A_t = a)$, through repeated application of the *Bellman operator* $T^\pi \colon \mathbb{R}^{\mathcal{S} \times \mathcal{A}} \to \mathbb{R}^{\mathcal{S} \times \mathcal{A}}$:

$$(T^\pi \hat{Q})(s, a) = \mathbb{E}_{(S', R') \sim \mathcal{T}(s,a)}[R' + \gamma \mathbb{E}_{A' \sim \pi(S')} \hat{Q}(S', A')]. \qquad (1)$$

Since $T^\pi$ is a contraction on $\mathbb{R}^{\mathcal{S} \times \mathcal{A}}$ with a unique fixed point $\hat{Q}^\pi$, that is $T^\pi \hat{Q}^\pi = \hat{Q}^\pi$, the iterated application of $T^\pi$ to any initial $\hat{Q} \in \mathbb{R}^{\mathcal{S} \times \mathcal{A}}$ yields $\hat{Q}^\pi$. The expectations in equation (1) can be estimated via Monte Carlo using experiences $(s, a, r, s')$ obtained through interaction with the MDP. A key challenge is then in obtaining experiences that are highly informative about the optimal policy.

A simple and effective approach to collecting such experiences is PSRL, a model-based algorithm based on two components: (i) a distribution over rewards and transition dynamics $P_{\hat{\mathcal{T}}}$ obtained using a Bayesian modelling approach, treating rewards and transition probabilities as random variables; and (ii) the *posterior sampling* exploration algorithm (Thompson, 1933; Dearden et al., 1998) which samples $\hat{\mathcal{T}} \sim P_{\hat{\mathcal{T}}}$, computes the optimal policy $\hat{\pi}$ with respect to the sampled $\hat{\mathcal{T}}$, and follows $\hat{\pi}$ for the duration of a single episode. The collected data are then used to update the $P_{\hat{\mathcal{T}}}$ model, and the whole process is iterated until convergence.

While PSRL performs very well on tabular problems, it is computationally expensive and does not utilise any additional information about the state space structure (e.g. visual similarity when states are represented by images). A family of methods called Randomised Value Functions (RVF; Osband et al., 2016b) attempt to overcome these issues by directly modelling a distribution over Q functions, $P_{\hat{Q}}$, instead of over MDPs, $P_{\hat{\mathcal{T}}}$. Rather than acting greedily with respect to a sampled MDP as in PSRL, the agent then acts greedily with respect to a sample $\hat{Q} \sim P_{\hat{Q}}$ drawn at the beginning of each episode, removing the main computational bottleneck. Since a parametric model is often chosen for $P_{\hat{Q}}$, the switch to Q function modelling also directly facilitates use of function approximation and thus generalisation between states.

## 3 Exploration under function approximation

Many exploration methods, including (Osband et al., 2016b,a; Moerland et al., 2017; O'Donoghue et al., 2018; Azizzadenesheli et al., 2018), can be interpreted as combining the concept of RVF with neural network function approximation. While the use of neural network function approximation allows these methods to scale to problems too complex for PSRL, it also brings about conceptual difficulties not present within PSRL and tabular RVF methods. Specifically, because a Q function is defined with respect to a particular policy, constructing $P_{\hat{Q}}$ requires selection of a reference policy or distribution over policies. Methods that utilise a distribution over reference policies typically

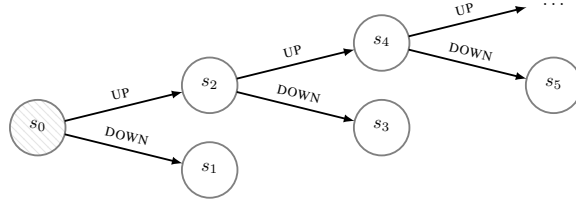

Figure 1: Binary tree MDP of size $L$. States $\mathcal{S} = \{s_0, \ldots, s_{2L}\}$ are one-hot encoded; actions $\mathcal{A} = \{a_1, a_2\}$ are mapped to movements $\{\text{UP}, \text{DOWN}\}$ according to a random mapping drawn independently for each state. Reward of one is obtained after reaching $s_{2L}$ and zero otherwise. States with odd indices and $s_{2L}$ are terminal.

employ a bootstrapped estimator of the Q function as we will discuss in more depth later. For now, we focus on methods that employ a single reference policy which commonly interleave two steps: (i) inference of $P_{\hat{Q}^{\pi_i}}$ for a given policy $\pi_i$ using the available data (*value prediction step*); (ii) estimation of an improved policy $\pi_{i+1}$ based on $P_{\hat{Q}^{\pi_i}}$ (*policy improvement step*). While a common policy improvement choice is $\pi_{i+1}\colon s \mapsto \mathbb{E}_{P_{\hat{Q}^{\pi_i}}}[G(\hat{Q})(s)]$, methods vary greatly in how they implement value prediction. To gain a better insight into the value prediction step, we examine its idealised implementation: Suppose we have access to a belief over MDPs, $P_{\hat{\mathcal{T}}}$ (as in PSRL), and want to compute the implied distribution $P_{\hat{Q}^{\pi}}$ for a *single* policy $\pi$. The intuitive (albeit still computationally expensive) procedure is to: (i) draw $\hat{\mathcal{T}} \sim P_{\hat{\mathcal{T}}}$; and (ii) repeatedly apply the Bellman operator $T^{\pi}$ to an initial $\hat{Q}$ for the drawn $\hat{\mathcal{T}}$ until convergence. Denoting by $F^{\pi}\colon \hat{\mathcal{T}} \mapsto \hat{Q}^{\pi}$ the map from $\hat{\mathcal{T}}$ to the corresponding $\hat{Q}^{\pi}$ for a policy $\pi$, the distribution of resulting samples is $P_{\hat{Q}^{\pi}} = F_{\#}^{\pi}P_{\hat{\mathcal{T}}}$.

This idealised value prediction step motivates, for example, the *Uncertainty Bellman Equation* (UBE; O'Donoghue et al., 2018). O'Donoghue et al. argue that to achieve effective exploration, it is necessary that the uncertainty about each $\hat{Q}^{\pi}(s, a)$, quantified by variance, is equal to the uncertainty about the immediate reward and the next state's Q value. This requirement can be formalised as follows:

**Definition 1** (Propagation of uncertainty). *For a given distribution $P_{\hat{\mathcal{T}}}$ and policy $\pi$, we say that a model $P_{\hat{Q}^{\pi}}$ propagates uncertainty according to $P_{\hat{\mathcal{T}}}$ if for each $(s, a) \in \mathcal{S} \times \mathcal{A}$ and $p = 1, 2$*

$$\mathbb{E}_{P_{\hat{Q}^{\pi}}}[\hat{Q}^{\pi}(s, a)^p] = \mathbb{E}_{F_{\#}^{\pi}P_{\hat{\mathcal{T}}}}[\hat{Q}^{\pi}(s, a)^p] = \mathbb{E}_{P_{\hat{\mathcal{T}}}}\big\{[\mathbb{E}_{(R', S') \sim \hat{\mathcal{T}}(s, a)}R' + \mathbb{E}_{A' \sim \pi(S')}F^{\pi}(\hat{\mathcal{T}})(S', A')]^p\big\}.$$

In words, *propagation of uncertainty* requires that the first two moments behave consistently under application of the Bellman operator.

Propagation of uncertainty is a desirable property when using *Upper Confidence Bounds* (UCB; Auer, 2002) for exploration, since UCB methods rely only on the first two moments of $P_{\hat{Q}^{\pi}}$. However, propagation of uncertainty is not sufficient for effective exploration under posterior sampling. We show this in the context of the binary tree MDP depicted in figure 1. To solve the MDP, the agent must execute a sequence of $L$ uninterrupted UP movements. In the following proposition, we show that any algorithm combining factorised symmetric distributions with posterior sampling (e.g. UBE) will solve this MDP with probability of at most $2^{-L}$ per episode, thus failing to outperform a uniform exploration policy. Importantly, the sizes of marginal variances have no bearing on this result, meaning that propagation of uncertainty on its own does not preclude this failure mode.

**Proposition 1.** *Let $|\mathcal{A}| > 1$, and $P_{\hat{Q}}$ be a factorised distribution, i.e. for $\hat{Q} \sim P_{\hat{Q}}$, $\hat{Q}(s, a)$ and $\hat{Q}(s', a')$ are independent, $\forall (s, a) \neq (s', a')$, with symmetric marginals. Assume that for each $s \in \mathcal{S}$, the marginal distributions of $\{\hat{Q}(s, a)\colon a \in \mathcal{A}\}$ are all symmetric around the same value $c_s \in \mathbb{R}$. Then the probability of executing any given sequence of $L$ actions under $\hat{\pi} \sim G_{\#}P_{\hat{Q}}$ is at most $2^{-L}$.*

Propagation of uncertainty is furthermore not necessary for posterior sampling. To see this, first note that for any given $P_{\hat{Q}^{\pi}}$, the posterior sampling procedure only depends on the induced distribution over greedy policies, i.e. the pushforward of $P_{\hat{Q}^{\pi}}$ by the greedy operator $G$. This means that from the point of view of posterior sampling, two Q function models are equivalent as long as they induce the same distribution over greedy policies. In what follows, we formalise this equivalence relationship (definition 2), and then show that each of the induced equivalence classes contains a model that does not propagate uncertainty (proposition 2), implying that posterior sampling does *not* rely on propagation of uncertainty.

**Definition 2** (Posterior sampling policy matching). *For a given distribution $P_{\hat{\mathcal{T}}}$ and a policy $\pi$, we say that a model $P_{\hat{Q}^\pi}$ matches the posterior sampling policy implied by $P_{\hat{\mathcal{T}}}$ if $G_\# P_{\hat{Q}^\pi} = (G \circ F^\pi)_\# P_{\hat{\mathcal{T}}}$.*

**Proposition 2.** *For any distribution $P_{\hat{\mathcal{T}}}$ and policy $\pi$ such that the variance $\mathbb{V}_{F_\#^\pi P_{\hat{\mathcal{T}}}}[\hat{Q}^\pi(s,a)]$ is greater than zero for some $(s,a)$, there exists a distribution $P_{\hat{Q}^\pi}$ which matches the posterior sampling policy (definition 2), but does not propagate uncertainty (definition 1), according to $P_{\hat{\mathcal{T}}}$.*

We conclude by addressing a potential criticism of proposition 1, i.e. that the described issues may be circumvented by initialising expected Q values to a value higher than the maximal attainable Q value in given MDP, an approach known as optimistic initialisation (Osband et al., 2016b). In such case, symmetries in the Q function may break as updates move the distribution towards more realistic Q values. However, when neural network function approximation is used, the effect of optimistic initialisation can disappear quickly with optimisation (Osband et al., 2018). In particular, with non-orthogonal state-action embeddings, Q value estimates may decrease for yet unseen state-action pairs, and estimates for different state-action states can move in tandem. In practice, most recent models employing neural network function approximation do not use optimistic initialisation (Osband et al., 2016a; Azizzadenesheli et al., 2018; Moerland et al., 2017; O'Donoghue et al., 2018).

## 4  Successor Uncertainties

We present *Successsor Uncertainties*, an algorithm which both propagates uncertainty and matches the posterior sampling policy. As our work is motivated by PSRL, we focus on the use with posterior sampling, leaving combination with other exploration algorithms for future research.

### 4.1  Q function model definition

Suppose we are given an embedding function $\phi\colon \mathcal{S}\times\mathcal{A}\to\mathbb{R}^d$, such that for all $(s,a)$, $\|\phi(s,a)\|_2 = 1$ and $\phi(s,a) \geq 0$ elementwise, and $\mathbb{E}_t R_{t+1} = \langle \phi_t, w \rangle$ for some $w \in \mathbb{R}^d$. Denote $\phi_t = \phi(S_t, A_t)$. Then we can express $Q_t^\pi$ as an inner product of $w$ and $\psi_t^\pi = \mathbb{E}_t[\sum_{\tau=t}^\infty \gamma^{\tau-t}\phi_\tau]$, the (discounted) expected future occurrence of each $\phi(s,a)$ feature under a policy $\pi$, as follows:

$$Q_t^\pi = \mathbb{E}_t \sum_{\tau=t}^\infty \gamma^{\tau-t} R_{\tau+1} = \mathbb{E}_t \sum_{\tau=t}^\infty \gamma^{\tau-t}\langle \phi_\tau, w\rangle = \left\langle \mathbb{E}_t \sum_{\tau=t}^\infty \gamma^{\tau-t}\phi_\tau, w \right\rangle = \langle \psi_t^\pi, w\rangle, \qquad (2)$$

where the second equality follows from the tower property of conditional expectation and the third from the dominated convergence theorem combined with the unit norm assumption.

The quantity $\psi_t^\pi$ is known in the literature as the *successor features* (Dayan, 1993; Barreto et al., 2017). Noting that $\psi_t^\pi = \phi_t + \gamma \mathbb{E}_t \psi_{t+1}^\pi$, an estimator of the successor features, $\hat{\psi}^\pi$, can be obtained by applying standard temporal difference learning techniques. The other quantity involved, $w$, can be estimated by regressing embeddings of observed states $\phi_t$ onto the corresponding rewards. We perform Bayesian linear regression to infer a distribution over rewards, using $\mathcal{N}(0, \theta I)$ as the prior over $w$ and $\mathcal{N}(\langle \phi, w\rangle, \beta)$ as the likelihood, which leads to posterior $\mathcal{N}(\mu_w, \Sigma_w)$ over $w$ with known analytical expressions for both $\mu_w$ and $\Sigma_w$. This induces posterior distribution over $\hat{Q}_{\text{SU}}^\pi$ given by

$$\hat{Q}_{\text{SU}}^\pi \sim \mathcal{N}(\hat{\Psi}^\pi \mu_w, \hat{\Psi}^\pi \Sigma_w (\hat{\Psi}^\pi)^\top), \qquad (3)$$

where $\hat{\Psi}^\pi = [\hat{\psi}^\pi(s,a)]_{(s,a)\in\mathcal{S}\times\mathcal{A}}^\top$. This is our Successor Uncertainties (SU) model for the Q function.

The final element of the SU model is the selection of a sequence of reference policies $(\pi_i)_{i\geq 1}$ for which the Q function model is learnt. We follow O'Donoghue et al. (2018) in constructing these iteratively as $\pi_{i+1}(s) = \mathbb{E}_{\hat{\pi}\sim G_\# P_{\hat{Q}^{\pi_i}}}[\hat{\pi}(s)]$.

### 4.2  Properties of the model

The non-diagonal covariance matrix of the SU Q function model (see equation (3)) means that SU does not suffer from the shortcomings of previous methods with factorised posterior distributions described in proposition 1. Moreover, note that $\hat{Q}_{\text{SU}}^\pi \sim F_\#^\pi P_{\hat{\mathcal{T}}}$ for the MDP model $P_{\hat{\mathcal{T}}}$ composed of a delta distribution concentrated on empirical transition frequencies, and the Bayesian linear model for rewards (assuming convergence of successor features, i.e. $\hat{\psi}^\pi = \psi^\pi$). SU thus both propagates uncertainty and matches the posterior sampling policy according to this choice of $P_{\hat{\mathcal{T}}}$.

However, due to its use of a point estimate for the transition probabilities, SU may underestimate Q function uncertainty, and a good model of transition probabilities which scales beyond tabular settings can lead to improved performance. Furthermore, SU estimates $P_{\hat{Q}^{\pi_{i+1}}}$ for a single policy, which we choose to be $\pi_{i+1}(s) = \mathbb{E}_{\hat{\pi} \sim G_\# P_{\hat{Q}^{\pi_i}}}[\hat{\pi}(s)]$. This approach may not adequately capture the uncertainty over $\hat{\pi}$ implied by $P_{\hat{Q}^{\pi_i}}$. We expect that incorporation of this uncertainty, or an improved method of choosing $\pi_{i+1}$, may further improve the SU algorithm.

## 4.3 Neural network function approximation

One of the main assumptions we made so far is that the embedding function $\phi$ is known a priori. This section considers the scenario where $\phi$ is to be estimated jointly with the other quantities using neural network function approximation. For reference, the pseudocode is included in appendix C.

Let $\hat{\phi} \colon \mathcal{S} \times \mathcal{A} \to \mathbb{R}^d_+$ be the current estimate of $\phi$, $(s_t, a_t)$ the state-action pair observed at step $t$, $r_{t+1}$ the reward observed after taking action $a_t$ in state $s_t$. Suppose we want to estimate the Q function of some given policy $\pi$, and denote $\hat{\phi}_t := \hat{\phi}(s_t, a_t)$, $\hat{\psi}_t := \hat{\psi}^\pi(s_t, a_t)$. We propose to jointly learn $\hat{\phi}$ and $\hat{\psi}$ by enforcing the known relationships between $\phi_t$, $\psi_t^\pi$ and $\mathbb{E}_t R_{t+1}$:

$$\min_{\hat{\phi},\hat{\psi},\hat{w}} \underbrace{\|\hat{\psi}_t - \hat{\phi}_t - \gamma(\hat{\psi}_{t+1})^\dagger\|_2^2}_{\text{successor feature loss}} + \underbrace{|\langle \hat{w}, \hat{\phi}_t \rangle - r_{t+1}|^2}_{\text{reward loss}} + \underbrace{|\langle \hat{w}, \hat{\psi}_t \rangle - \gamma(\langle \hat{w}, \hat{\psi}_{t+1} \rangle)^\dagger - r_{t+1}|^2}_{\text{Q value loss}} \quad (4)$$

in expectation over the observed data $\{(s_t, a_t, r_{t+1} s_{t+1}) \colon t = 0, \dots, N\}$ with $a_{t+1} \sim \pi(s_{t+1})$; $\hat{\phi}_t, \hat{\psi}_t \in \mathbb{R}^d_+, \|\hat{\phi}_t\|_2 = 1, \forall t$, are respectively ensured by the use of ReLU activations and explicit normalisation. The $\hat{w} \in \mathbb{R}^d$ are the final layer weights shared by the the reward and the Q value networks. Quantities superscripted with $\dagger$ are treated as fixed during optimisation.

The need for the successor feature and reward losses follows directly from the definition of the SU model. We add the explicit Q value loss to ensure accuracy of Q value predictions. Assuming that there exists a (ReLU) network that achieves zero successor feature and reward loss, the added Q value loss has no effect. However, finding such an optimal solution is difficult in practice and empirically the addition of the Q value loss improves performance. Our modelling assumptions cause all constituent losses in equation (4) to have similar scale, and thus we found it unnecessary to introduce weighting factors. Furthermore, unlike in previous work utilising successor features (Kulkarni et al., 2016; Machado et al., 2017, 2018), SU does not rely on any auxiliary state reconstruction or state-transition prediction tasks for learning, which simplifies implementation and greatly reduces the required amount of computation.

We employ the neural network output weights $\hat{w}$ in prediction of the mean $Q$ function, and use the Bayesian linear model only to provide uncertainty estimates. In estimating the covariance matrix $\Sigma_w$, we decay the contribution of old data-points, $\hat{\Sigma}_w = (\zeta^N \theta^{-1} I + \beta^{-1} \sum_{i=0}^N \zeta^{N-i} \hat{\phi}_i \hat{\phi}_i^\top)^{-1}, \zeta \in [0,1]$, so as to counter non-stationarity of the learnt state-action embeddings $\hat{\phi}$.

## 4.4 Comparison to existing methods

We discuss two popular classes of Q function models compatible with neural network function approximation: methods relying on Bayesian linear Q function models and methods based on bootstrapping. We omit variational Q-learning methods such as (Gal, 2016; Lipton et al., 2018), as conceptual issues with these algorithms have already been identified in an illuminating line of work by Osband et al. (2016a, 2018).

Bayesian linear Q function models encompass our SU algorithm, UBE (O'Donoghue et al., 2018) implemented with value function approximation, Bayesian Deep Q Networks (BDQN; Azizzadenesheli et al., 2018), and a range of other related work (Levine et al., 2017; Moerland et al., 2017). The algorithms within this category tend to use a Q function model of the form $\hat{Q}^\pi(s, a) = \langle \hat{\phi}_s^\pi, w_a \rangle$, where $\hat{\phi}_s^\pi$ are state embeddings and $w_a \sim P_{w_a}$ are weights of a Bayesian linear model. The embeddings $\hat{\phi}_s^\pi$ are produced by a neural network, and are usually optimised using a temporal difference algorithm applied to Q values. However, these methods do not enforce any explicit structure within the embeddings $\hat{\phi}_s^\pi$ which would be required for posterior sampling policy matching, and prevent these methods from falling victim to proposition 1. SU can thus be viewed as a simple and computationally cheap alternative fixing the issues of existing Bayesian linear Q function models.

Bootstrapped DQN (Osband et al., 2016a, 2018) is a model which consists of an ensemble of $K$ standard Q networks, each initialised independently and trained on a random subset of the observed

data. Each network is augmented with a fixed additive prior network, so as to ensure the ensemble distribution does not collapse in sparse environments. If all networks within the ensemble are trained to estimate the Q function for a single policy $\pi$, then Bootstrapped DQN both propagates uncertainty and matches the posterior sampling policy for a distribution over MDPs formed by the mixture over empirical MDPs corresponding to each subsample of the data. In practice, Bootstrapped DQN does not assume a single policy $\pi$ and instead each network learns for its corresponding greedy policy. Bootstrapped DQN is, however, more computationally expensive: its performance increases with the size of the ensemble $K$, but so does the amount of computation required. Our experiments show that SU is much cheaper computationally, and that despite using only a single reference policy, it manages to outperform Bootstrapped DQN on a wide range of exploration tasks (see section 5).

## 5    Tabular experiments

We present results for: (i) the binary tree MDP accompanied by theoretical analysis showing how SU succeeds and avoids the pitfalls identified in proposition 1; (ii) a hard exploration task proposed by Osband et al. (2018) together with the Boostrapped DQN algorithm which SU outperforms by a significant margin.[3] We also provide an analysis explaining why some of the previously discussed algorithms perform well on seemingly similar experiments present in existing literature.

### 5.1    Binary tree MDP

We study the behaviour of SU and its competitors on the binary tree MDP introduced in figure 1. Figure 2 shows the empirical performance of each algorithm as a function of the tree size $L$. Evidently, both BDQN and UBE fail to outperform a uniform exploration policy. For UBE, this is a consequence of proposition 1, and the similarly poor behaviour of BDQN suggests it may suffer from an analogous issue. In contrast, SU and Bootstrapped DQN are able to succeed on large binary trees despite the very sparse reward structure and randomised action effects. However, Bootstrapped DQN requires approximately 25 times more computation than SU to approach similar levels of performance due to the necessity to train a whole ensemble of Q networks.

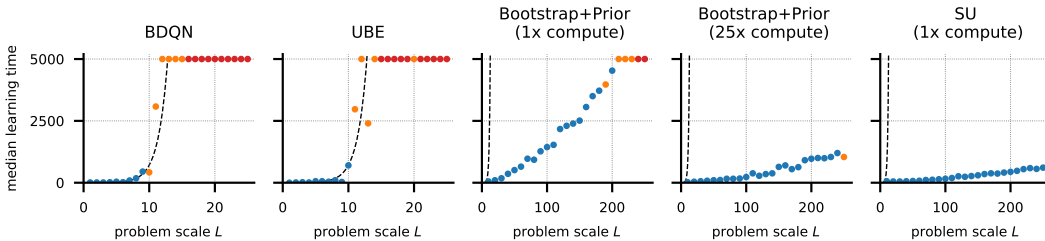

Figure 2: Median number of episodes required to learn the optimal policy on the tree MDP. Blue points indicate all 5 seeds succeeded within 5000 episodes, orange indicates only some of the runs succeeded, and red all runs failed. Dashed lines correspond to the median for a uniform exploration policy. Note the reduced size of the x-axis for BDQN and UBE.

The next proposition and its proof provide intuition for the success of SU on the tree MDP. The proof is based on a lemma stated just after the proposition (see appendix B.1 for formal treatment).

**Proposition 3** (Informal statement). *Assume the SU model with: (i) fixed one-hot state-action embeddings $\phi$, (ii) uniform exploration thus far, (iii) successor representations learnt to convergence for a uniform policy. Let $s_k$ for $2 \leq k < 2L$, even, be a state visited $N$ times thus far. Then the probability of selecting* UP *in $s_k$, given* UP *was selected in $s_0, s_2, \ldots, s_{k-2}$, is greater than one half with probability greater than $1 - \epsilon_N$, where $\epsilon_N$ decreases exponentially with $N$.*

**Lemma 4** (Informal statement). *Under the SU model $\hat{Q} \sim P_{\hat{Q}^\pi}$ for the uniform policy $\pi$, the probability that the greedy policy $\hat{\pi} = G(\hat{Q})$ selects* UP *in $s_k$, given* UP *was selected in $s_0, s_2, \ldots, s_{k-2}$, is greater than one half if there exists an even $0 \leq j < k$ such that*

$$\mathrm{Cov}(\hat{Q}(s_k, \mathrm{UP}), \hat{Q}(s_j, \mathrm{UP})) > \mathrm{Cov}(\hat{Q}(s_k, \mathrm{DOWN}), \hat{Q}(s_j, \mathrm{UP})).$$

*Sketch proof of proposition 3.* Under SU $\hat{Q}(s_j, \text{UP}) = \hat{r}(s_j, \text{UP}) + \ldots + \rho\hat{Q}(s_k, \text{UP}) + \rho\hat{Q}(s_k, \text{DOWN})$ with $\rho = 2^{-\left(\frac{k-j}{2}\right)}$ the probability of getting from $s_j$ to $s_k$ under the uniform policy. Note that $\hat{Q}(s_j, \text{UP})$ and $\hat{Q}(s_k, \text{DOWN})$ only share the $\hat{Q}(s_k, \text{DOWN}) = \hat{r}(s_k, \text{DOWN})$ term, whereas $\hat{Q}(s_k, \text{UP})$ and $\hat{Q}(s_j, \text{UP})$ share $\hat{r}(s_j, \text{UP}), \ldots, \hat{r}(s_p, \text{DOWN})$, where $s_p$ is the state with the highest index seen so far. Thus covariance between $\hat{Q}(s_k, \text{UP})$ and $\hat{Q}(s_j, \text{UP})$ is higher than that between $\hat{Q}(s_k, \text{DOWN})$ and $\hat{Q}(s_j, \text{UP})$ with high probability (at least $1 - \epsilon_N$), and the result follows from lemma 4. □

Proposition 3 implies that (at least under the simplifying assumption of prior exploration being uniform) SU is likely to assign higher probability to Q functions for which a greedy policy leads towards the furthest visited state (cf. the role of the state $s_p$ in the sketch proof). This is a strategy actively aimed for in exploration algorithms such as Go-Explore where the agent uses imitation learning to return to the furthest discovered states (Ecoffet et al., 2019).

## 5.2   Chain MDP from (Osband et al., 2018)

We present results on the chain environment introduced by Osband et al. (2018), described in detail in appendix C.1. Osband et al. describe their MDP as being "akin to looking for a piece of hay in a needle-stack" and state that it "may seem like an impossible task". Figure 3 shows the scaling for Successor Uncertainties and Bootstrap+Prior for this problem. Learning time $T$ scales empirically as $\mathcal{O}(L^{2.5})$ for SU, versus $\mathcal{O}(L^3)$ for Bootstrap+Prior (as reported in Osband et al., 2018).

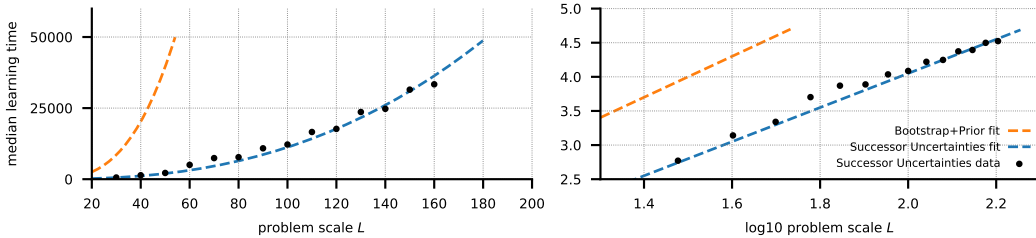

Figure 3: Learning time $T$ for SU and Bootstrap+Prior for a range of problem sizes $L$ on the chain MDP. Curve for SU is $\log_{10} T = 2.5 \log_{10} L - 0.95$. Curve for Bootstrap+Prior is taken from figure 8 in (Osband et al., 2018).

## 5.3   On the success of BDQN in environments with tied actions

We briefly address prior results in the literature where BDQN is seen solving problems seemingly similar to our binary tree MDP with ease (as in, for example, figure 1 of Touati et al., 2018). The discrepancy occurs because previous work often does not randomise the effects of actions (for example Osband et al., 2016a; Plappert et al., 2018; Touati et al., 2018), i.e. if $a_1$ leads UP in any state $s_k$, then $a_1$ leads UP in all states. We refer to this as the *tied actions* setting. In the following proposition, we show that MDPs with tied actions are trivial for BDQN with strictly positive activations (e.g. sigmoid). We offer a similar result for ReLU in appendix B.2.

**Proposition 5.** *Let $\hat{Q}(s, a) = \langle \phi(s), w_a \rangle$ be a Bayesian Q function model with $\phi(s) = \varphi(U1_s) \in \mathbb{R}^d$, $1_s$ a one-hot encoding of $s$, and $\varphi$ a strictly positive activation function (e.g. sigmoid) applied elementwise. Then sampling independently from the prior $w_a \sim \mathcal{N}(0, \sigma_w^2 I)$, $U_{hs} \sim \mathcal{N}(0, \sigma_u^2)$ solves a tied action binary tree of size $L$ in $T \leq -[\log_2(1 - 2^{-d})]^{-1}$ median number of episodes.*

*Proof.* Define $\Delta := w_{\text{UP}} - w_{\text{DOWN}}$ and observe UP is selected if $\hat{Q}(s, \text{UP}) - \hat{Q}(s, \text{DOWN}) = \langle \phi(s), w_{\text{UP}} - w_{\text{DOWN}} \rangle > 0$. By strict positivity of $\varphi$, the probability that UP is always selected

$$\mathbb{P}\Big[\bigcap_{j=0}^{L-1} \{\hat{Q}(s_{2j}, \text{UP}) > \hat{Q}(s_{2j}, \text{DOWN})\}\Big] \geq \mathbb{P}\Big[\bigcap_{j=0}^{L-1} \{\langle \phi(s_{2j}), \Delta \rangle > 0\} \mid \Delta > 0\Big] \mathbb{P}(\Delta > 0) = \mathbb{P}(\Delta > 0),$$

where $\Delta > 0$ is to be interpreted elementwise. As $\Delta \sim \mathcal{N}(0, 2\sigma_w^2 I)$, $\mathbb{P}(\Delta > 0) = 2^{-d}$ for all $L$. □

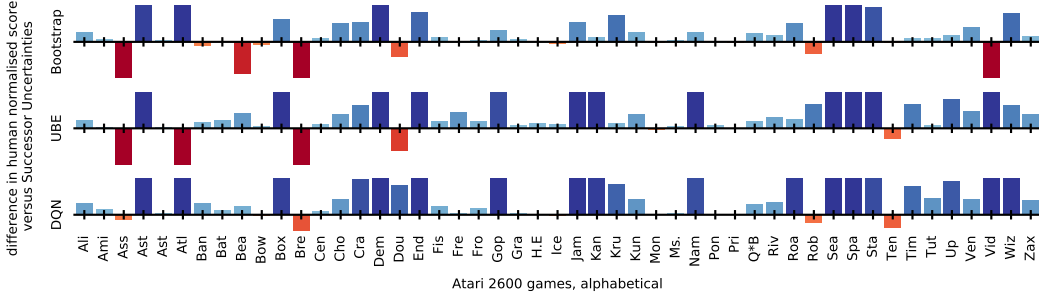

Figure 4: Bars show the difference in human normalised score between SU and Bootstrap DQN (top), UBE (middle) and DQN (bottom) for each of the 49 Atari 2600 games. Blue indicates SU performed better, red worse. SU outperforms the baselines on $36/49$, $43/49$ and $42/49$ games respectively. Y-axis values have been clipped to $[-2.5, 2.5]$.

A single layer BDQN with one neuron can thus solve a tied action binary tree of any size $L$ in one episode (median) while completely ignoring all state information. That such an approach can be successful implies tied actions MDPs generally do not make for good exploration benchmarks.

## 6   Atari 2600 experiments

We have tested the SU algorithm on the standard set of 49 games from the Arcade Learning Environment, with the aim of showing that SU can be scaled to complex domains that require generalisation between states. We use a standard network architecture as in (Mnih et al., 2015; Van Hasselt et al., 2016) endowed with an extra head for prediction of $\hat{\phi}$ and one-step value updates. More detail on our implementation, network architecture and training procedure can be found in appendix C.2.[4]

SU obtains a median human normalised score of 2.09 (averaged over 3 seeds) after 200M training frames under the 'no-ops start 30 minute emulator time' test protocol described in (Hessel et al., 2018). Table 1 shows we significantly outperform competing methods. The raw scores are reported in table 2 (appendix), and the difference in human normalised score between SU and the competing algorithms for individual games is charted in figure 4. Since Azizzadenesheli et al. (2018) only report scores for a small subset of the games and use a non-standard testing procedure, we do not compare against BDQN. Osband et al. (2018), who introduce Bootstrap+Prior, do not report Atari results; we thus compare with results for the original plain Bootstrapped DQN (Osband et al., 2016a) instead.

Table 1: Human normalised Atari scores. Superhuman performance is the percentage of games on which each algorithm surpasses human performance (as reported in Mnih et al., 2015).

| Algorithm | Human normalised score percentiles | | | Superhuman |
|---|---|---|---|---|
| | 25% | 50% | 75% | performance % |
| Successor Uncertainties | **1.06** | **2.09** | **5.95** | **77.55%** |
| Bootstrapped DQN | 0.76 | 1.60 | 5.16 | 67.35% |
| UBE | 0.38 | 1.07 | 4.14 | 51.02% |
| DQN + $\epsilon$-greedy | 0.50 | 1.00 | 3.41 | 48.98% |

## 7   Conclusion

We studied the Posterior Sampling for Reinforcement Learning algorithm and its extensions within the Randomised Value Function framework, focusing on use with neural network function approximation. We have shown theoretically that exploration techniques based on the concept of propagation of uncertainty are neither sufficient nor necessary for posterior sampling exploration in sparse environments. We instead proposed posterior sampling policy matching, a property motivated by the probabilistic model over rewards and state transitions within the PSRL algorithm. Based on the theoretical insights,

we developed Successor Uncertainties, a randomised value function algorithm that avoids some of the pathologies present within previous work. We showed empirically that on hard tabular examples, SU significantly outperforms competing methods, and provided theoretical analysis of its behaviour. On Atari 2600, we demonstrated Successor Uncertainties is also highly effective when combined with neural network function approximation.

Performance on the hardest exploration tasks often benefits greatly from multi-step temporal difference learning (Precup, 2000; Munos et al., 2016; O'Donoghue et al., 2018) which we believe is the most promising direction for improving Successor Uncertainties. Since modification of existing models to incorporate Successor Uncertainties is relatively simple, other standard techniques used within model-free reinforcement learning like (Schaul et al., 2015; Wang et al., 2016) can be leveraged to obtain further gains. This paper thus opens many exciting directions for future research which we hope will translate into both further performance improvements and a more thorough understanding of exploration in modern reinforcement learning.

### Acknowledgements

We thank Matej Balog and the anonymous reviewers for their helpful comments and suggestions. Jiri Hron acknowledges support by a Nokia CASE Studentship.

## Footnotes

[3]Code for the tabular experiments: `https://djanz.org/successor_uncertainties/tabular_code`

[4]Code for the Atari experiments: `djanz.org/successor_uncertainties/atari_code`

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
