[Supplementary Material · successor_uncertainties_appendices.pdf]

Table 2: Raw scores for Successor Uncertainties alongside DQN, UBE and Bootstrap DQN . Test conditions: 30 minute emulator time limit and no-ops starting condition. Baselines as reported in (Hessel et al., 2018).

| Game | DQN | UBE | Bootstrap DQN | SU |
|---|---|---|---|---|
| Alien | 1,620.0 | 3,345.3 | 2,436.6 | **6,924.4** |
| Amidar | 978.0 | 1,400.1 | 1,272.5 | **1,574.4** |
| Assault | 4,280.4 | **11,521.5** | 8,047.1 | 3,813.8 |
| Asterix | 4,359.0 | 7,038.5 | 19,713.2 | **42,762.2** |
| Asteroids | 1,364.5 | 1,159.4 | 1,032.0 | **2,270.4** |
| Atlantis | 279,987.0 | **4,648,770.8** | 994,500.0 | 2,026,261.1 |
| Bank Heist | 455.0 | 718.0 | **1,208.0** | 1,017.4 |
| Battle Zone | 29,900.0 | 19,948.9 | 38,666.7 | **39,944.4** |
| Beam Rider | 8,627.5 | 6,142.4 | **23,429.8** | 11,652.3 |
| Bowling | 50.4 | 18.3 | **60.2** | 38.3 |
| Boxing | 88.0 | 34.2 | 93.2 | **99.7** |
| Breakout | 385.5 | 617.3 | **855.0** | 352.7 |
| Centipede | 4,657.7 | 4,324.1 | 4,553.5 | **7,049.3** |
| Chopper Command | 6,126.0 | 7,130.8 | 4,100.0 | **15,787.8** |
| Crazy Climber | 110,763.0 | 132,997.5 | 137,925.9 | **171,991.1** |
| Demon Attack | 12,149.4 | 25,021.1 | 82,610.0 | **183,243.2** |
| Double Dunk | -6.6 | **4.7** | 3.0 | -0.2 |
| Enduro | 729.0 | 30.8 | 1,591.0 | **2,216.3** |
| Fishing Derby | -4.9 | 3.1 | 26.0 | **53.3** |
| Freeway | 30.8 | 0.0 | **33.9** | 33.8 |
| Frostbite | 797.4 | 546.0 | 2,181.4 | **2,733.3** |
| Gopher | 8,777.4 | 13,808.0 | 17,438.4 | **19,126.2** |
| Gravitar | 473.0 | 224.5 | 286.1 | **684.4** |
| H.E.R.O. | 20,437.8 | 12,808.8 | 21,021.3 | **22,050.8** |
| Ice Hockey | -1.9 | -6.6 | **-1.3** | -2.9 |
| James Bond | 768.5 | 778.4 | 1,663.5 | **2,171.1** |
| Kangaroo | 7,259.0 | 6,101.2 | 14,862.5 | **15,751.1** |
| Krull | 8,422.3 | 9,835.9 | 8,627.9 | **10,103.9** |
| Kung-Fu Master | 26,059.0 | 29,097.1 | 36,733.3 | **50,878.9** |
| Montezumas Revenge | 0.0 | **499.1** | 100.0 | 0.0 |
| Ms. Pac-Man | 3,085.6 | 3,141.3 | 2,983.3 | **4,894.8** |
| Name This Game | 8,207.8 | 4,604.4 | 11,501.1 | **12,686.7** |
| Pong | 19.5 | 14.2 | 20.9 | **21.0** |
| Private Eye | 146.7 | -281.1 | **1,812.5** | 133.3 |
| Q*Bert | 13,117.3 | 16,772.5 | 15,092.7 | **22,895.8** |
| River Raid | 7,377.6 | 8,732.3 | 12,845.0 | **17,940.6** |
| Road Runner | 39,544.0 | 56,581.1 | 51,500.0 | **61,594.4** |
| Robotank | 63.9 | 42.4 | **66.6** | 58.5 |
| Seaquest | 5,860.6 | 1,880.6 | 9,083.1 | **68,739.9** |
| Space Invaders | 1,692.3 | 2,032.4 | 2,893.0 | **13,754.3** |
| Star Gunner | 54,282.0 | 44,458.6 | 55,725.0 | **78,837.8** |
| Tennis | **12.2** | 10.2 | 0.0 | -1.0 |
| Time Pilot | 4,870.0 | 5,650.6 | 9,079.4 | **9,574.4** |
| Tutankham | 68.1 | 218.6 | 214.8 | **247.7** |
| Up and Down | 9,989.9 | 12,445.9 | 26,231.0 | **29,993.4** |
| Venture | 163.0 | -14.7 | 212.5 | **1,422.2** |
| Video Pinball | 196,760.4 | 51,178.2 | **811,610.0** | 515,601.9 |
| Wizard Of Wor | 2,704.0 | 8,425.5 | 6,804.7 | **15,023.3** |
| Zaxxon | 5,363.0 | 5,717.9 | 11,491.7 | **14,757.8** |

# A Appendix to section 3: proofs of propositions 1 and 2

**Proposition 1.** *Let $|\mathcal{A}| > 1$, and $P_{\hat{Q}}$ be a factorised distribution, i.e. for $\hat{Q} \sim P_{\hat{Q}}$, $\hat{Q}(s,a)$ and $\hat{Q}(s',a')$ are independent, $\forall (s,a) \neq (s',a')$, with symmetric marginals. Assume that for each $s \in \mathcal{S}$, the marginal distributions of $\{\hat{Q}(s,a) \colon a \in \mathcal{A}\}$ are all symmetric around the same value $c_s \in \mathbb{R}$. Then the probability of executing any given sequence of $L$ actions under $\hat{\pi} \sim G_\# P_{\hat{Q}}$ is at most $2^{-L}$.*

*Proof.* We can w.l.o.g. assume that the distribution is symmetric around zero as centring will not affect validity of the following argument. To attain probability of taking a particular action $a$ in state $s$ greater than $\frac{1}{2}$, it must be that $\mathbb{P}(a = \operatorname{argmax}_{a'} \hat{Q}(s,a')) > \frac{1}{2}$. This event can be described as

$$A \coloneqq \bigcap_{a' \in \mathcal{A} \backslash \{a\}} \{\hat{Q} \colon \hat{Q}(s,a) > \hat{Q}(s,a')\};$$

by symmetry, the event

$$\tilde{A} \coloneqq \bigcap_{a' \in \mathcal{A} \backslash \{a\}} \{\hat{Q} \colon \hat{Q}(s,a) < \hat{Q}(s,a')\},$$

must have the same probability as $A$. Because $\mathbb{P}(A) + \mathbb{P}(\tilde{A}) \leq 1$, it must be that $\mathbb{P}(A) \leq \frac{1}{2}$. Since $\hat{Q}(s,a)$ is by assumption independent of any $\hat{Q}(s',a')$, $(s,a) \neq (s',a')$, the probability of executing a sequence of $L$ actions is at best (i.e. under deterministic transitions) the product of probabilities of executing a single action, which is upper bounded by $2^{-L}$. $\square$

**Proposition 2.** *For any distribution $P_{\hat{\mathcal{T}}}$ and policy $\pi$ such that the variance $\mathbb{V}_{F_\#^\pi P_{\hat{\mathcal{T}}}}[\hat{Q}^\pi(s,a)]$ is greater than zero for some $(s,a)$, there exists a distribution $P_{\hat{Q}^\pi}$ which matches the posterior sampling policy (definition 2), but does not propagate uncertainty (definition 1), according to $P_{\hat{\mathcal{T}}}$.*

*Proof.* First, let us formally define $G \colon \bar{\mathbb{R}}^{\mathcal{S} \times \mathcal{A}} \to \mathcal{A}^{\mathcal{S}}$ to be the function which maps each Q function to the corresponding greedy policy (we can w.l.o.g. assume there is some tie-breaking rule for when $\hat{Q}(s,a) = \hat{Q}(s,a'), a \neq a'$, e.g. taking the action with smaller index). Here, $\bar{\mathbb{R}}$ is the extended space of real numbers, and we assume the Borel $\sigma$-algebra generated by the usual interval topology; the discrete $\sigma$-algebra is assumed on $\mathcal{A}$. For product spaces, the product $\sigma$-algebra is taken. Given that the pre-image of a particular point $\hat{\pi} \in \mathcal{A}^{\mathcal{S}}$ is $\bigcap_{s \in \mathcal{S}} \{\hat{Q} \colon \hat{Q}(s, \hat{\pi}(s)) \geq \hat{Q}(s,a), \forall a\}$, $G$ is measurable and thus the distribution $P_{\hat{\pi}} = G_\# P_{\hat{Q}}$ is well-defined for any $P_{\hat{Q}} \in \mathcal{P}(\mathbb{R}^{\mathcal{S} \times \mathcal{A}})$, and in particular for $P_{\hat{Q}} = (G \circ F^\pi)_\# P_{\hat{\mathcal{T}}}$ for any policy $\pi$.

Our proof relies on the following observation: if we sample $\hat{\pi} \sim P_{\hat{\pi}}$ and then use it to explore the environment, the distribution of actions taken in a particular state $s \in \mathcal{S}$ will be categorical with parameter $p_s \in \{p \in \mathbb{R}_+^{|\mathcal{A}|} \colon \sum_{j=1}^{|\mathcal{A}|} p_j = 1\}$ (except for when the state $s$ is reached with probability zero under $P_{\hat{\mathcal{T}}}$ and $P_{\hat{\pi}}$ in which case we can set $p_s$, for example, to $[1/|\mathcal{A}|, \ldots, 1/|\mathcal{A}|]^\top$ as this will not affect the following argument). Hence to achieve $G_\# P_{\hat{Q}^\pi} = P_{\hat{\pi}}$, it is sufficient to construct a model $\hat{Q} \sim P_{\hat{Q}^\pi}$ for which the distribution of $\operatorname{argmax}_{a \in \mathcal{A}} \hat{Q}(s,a)$ is categorical with the parameter $p_s$ for all $s \in \mathcal{S}$. We achieve this using the Gumbel trick: sample $g_{sa} \sim \operatorname{Gumbel}(0,1)$ independently for each $(s,a) \in \mathcal{S} \times \mathcal{A}$, and set $\hat{Q}(s,a) = g_{sa} + \log p_{sa}$ (interpreting $\log 0 = -\infty$).

To finish the proof, observe that if the inputs to the $\operatorname{argmax}$ operator are all shifted by the same amount, or multiplied by a positive scalar, the output remains unchanged. Hence taking $\hat{Q}'(s,a) = a + b\hat{Q}(s,a)$ for any $a \in \mathbb{R}, b > 0$ will also result in the desired distribution over exploration policies. We can thus take the $(s,a)$ for which $\mathbb{V}_{F_\#^\pi P_{\hat{\mathcal{T}}}}[\hat{Q}(s,a)] > 0$ and pick $b > 0$ so that $\mathbb{V}_{P_{\hat{Q}^\pi}}[\hat{Q}(s,a)] \neq \mathbb{V}_{F_\#^\pi P_{\hat{\mathcal{T}}}}[\hat{Q}(s,a)]$ which will be always possible as $\mathbb{V}(b\hat{Q}(s,a))$ is $b^2 \mathbb{V}(g_{sa}) = b^2 \frac{\pi^2}{6}$ if $p_{sa} > 0$ and is undefined otherwise. $\square$

# B Appendix to section 5

## B.1 Proofs for section 5.1

In what follows, the binary tree MDP of size $L$ introduced in figure 1 is assumed. We further assume $\phi$ is given and maps each state-action to its one-hot embedding. As all of the following arguments are independent of the mapping from the actions $\{a_1, a_2\}$ to the movements $\{\text{UP}, \text{DOWN}\}$, we use $\mathcal{A} = \{\text{UP}, \text{DOWN}\}$ directly for improved clarity.

To prove lemma 4, we will need lemmas 6 to 9 which we state and prove now.

**Lemma 6.** *After any number of posterior updates, the SU reward distribution is multivariate normal with all rewards mutually independent. Furthemore, under the SU Q function model $\hat{Q} \sim P_{\hat{Q}^\pi}$ for any policy $\pi$, and even state indices $0 \leq j < k$*

$$\text{Cov}(\hat{Q}(s_k, \text{UP}), \hat{Q}(s_j, \text{DOWN})) = \text{Cov}(\hat{Q}(s_k, \text{DOWN}), \hat{Q}(s_j, \text{DOWN})) = 0\,.$$

*Proof.* Inspecting equations (2) and (3), it is easy to see that neither $\hat{Q}(s_k, \text{UP})$ and $\hat{Q}(s_j, \text{DOWN})$ nor $\hat{Q}(s_k, \text{DOWN})$ and $\hat{Q}(s_j, \text{DOWN})$ share any reward terms, since $j < k$ by assumption and the empirical transition frequencies used to construct $P_{\hat{Q}^\pi}$ will always be zero if the true transition probability is zero (recall that DOWN always terminates the episode). Hence assuming that the successor features were successfully learnt, i.e. $\hat{\psi}^\pi = \psi^\pi$, it is sufficient to show that the individual rewards are independent for SU. To see that this is the case, observe that the assumed one-hot encoding of state-actions implies that SU reward distribution will be a multivariate Gaussian with diagonal covariance after any number of updates which implies the desired independence. □

**Lemma 7.** *Under the SU model $\hat{Q} \sim P_{\hat{Q}^\pi}$ for any policy $\pi$, the random vector $\Delta$, $\Delta_{k/2} \coloneqq \hat{Q}(s_k, \text{UP}) - \hat{Q}(s_k, \text{DOWN})$, follows a zero mean Gaussian distribution with $\text{Cov}(\Delta_{k/2}, \Delta_{j/2}) = \text{Cov}(\hat{Q}(s_k, \text{UP}), \hat{Q}(s_j, \text{UP})) - \text{Cov}(\hat{Q}(s_k, \text{DOWN}), \hat{Q}(s_j, \text{UP})))$ for any even indices $0 \leq j < k$.*

*Proof.* The Gaussianity of the joint distribution of $\Delta_{j/2}$ and $\Delta_{k/2}$ follows from the linearity property of multivariate normal distributions. For the covariance, observe

$$\begin{aligned}
\text{Cov}(\Delta_{k/2}, \Delta_{j/2}) &= \text{Cov}(\hat{Q}(s_k, \text{UP}) - \hat{Q}(s_k, \text{DOWN}), \hat{Q}(s_j, \text{UP}) - \hat{Q}(s_j, \text{DOWN})) \\
&= \text{Cov}(\hat{Q}(s_k, \text{UP}), \hat{Q}(s_j, \text{UP})) - \text{Cov}(\hat{Q}(s_k, \text{DOWN}), \hat{Q}(s_j, \text{UP})) - \\
&\quad \text{Cov}(\hat{Q}(s_k, \text{UP}), \hat{Q}(s_j, \text{DOWN})) + \text{Cov}(\hat{Q}(s_k, \text{DOWN}), \hat{Q}(s_j, \text{DOWN})) \\
&= \text{Cov}(\hat{Q}(s_k, \text{UP}), \hat{Q}(s_j, \text{UP})) - \text{Cov}(\hat{Q}(s_k, \text{DOWN}), \hat{Q}(s_j, \text{UP})))\,,
\end{aligned}$$

where we used bilinearity of the covariance operator and then applied lemma 6. □

**Lemma 8.** *Under the SU model $\hat{Q} \sim P_{\hat{Q}^\pi}$ for the uniform policy $\pi$, and even indices $0 \leq j < k$*

$$\text{Cov}(\hat{Q}(s_k, \text{UP}), \hat{Q}(s_j, \text{UP})) > \text{Cov}(\hat{Q}(s_k, \text{DOWN}), \hat{Q}(s_j, \text{UP})))$$
$$\iff \qquad \mathbb{V}(\hat{Q}(s_k, \text{UP})) > \mathbb{V}(\hat{Q}(s_k, \text{DOWN}))\,.$$

*Proof.* Analogously to the proof of lemma 7, we see that under the uniform policy

$$\begin{aligned}
&\text{Cov}(\hat{Q}(s_k, \text{UP}), \hat{Q}(s_j, \text{UP})) \\
=\quad &\text{Cov}(\hat{Q}(s_k, \text{UP}), \hat{r}(s_j, \text{UP}) + 2^{-1}\hat{r}(s_{j+2}, \text{UP}) + \ldots + 2^{-(\frac{k-j}{2})}\hat{Q}(s_k, \text{UP})) \\
=\quad &2^{-(\frac{k-j}{2})}\,\text{Cov}(\hat{Q}(s_k, \text{UP}), \hat{Q}(s_k, \text{UP})) = 2^{-(\frac{k-j}{2})}\,\mathbb{V}(\hat{Q}(s_k, \text{UP}))\,,
\end{aligned}$$

where the $2^{-l}$ terms correspond to the probability of getting to $s_l$ from $(s_j, \text{UP})$, $l = 1, 2, \ldots, \frac{k-j}{2}$, and we used bilinearity of the covariance operator and then applied lemma 6. An analogous argument yields $\text{Cov}(\hat{Q}(s_k, \text{DOWN}), \hat{Q}(s_j, \text{UP})) = 2^{-(\frac{k-j}{2})}\mathbb{V}(\hat{Q}(s_k, \text{DOWN}))$, concluding the proof. □

**Lemma 9.** *For a d-dimensional centred Gaussian random vector $\Delta \sim \mathcal{N}(0, \Sigma)$ with $\text{Cov}(\Delta_d, \Delta_i) > 0$ for all $i = 1, \ldots, d-1$, the following bound holds: $\mathbb{P}(\Delta_d > 0 \mid \Delta_1 > 0, \ldots, \Delta_{d-1} > 0) > 1/2$.*

*Proof.* Notice that $\Delta$ and $\Sigma^{1/2}X$, $X \sim \mathcal{N}(0,1)$, are equal in distribution which allows us to set $\Delta_i = \langle v_i, X \rangle$, with $v_i \in \mathbb{R}^d$ the $i^{\text{th}}$ row of $\Sigma^{1/2}$. Let $R_v \colon \mathbb{R}^d \to \mathbb{R}^d$ be the reflection against the orthogonal complement of $v$, i.e.

$$R_v(x) = x - 2\frac{\langle x, v \rangle}{\langle v, v \rangle}v \,.$$

It is easy to see that $\langle v, R_v(x) \rangle = -\langle v, x \rangle$ and consequently $R_v(R_v(x)) = x$. The main idea of this proof is to partition $\mathbb{R}^d$ into the half-spaces $\{x \colon \langle v_i, x \rangle > 0\}$ and $\{x \colon \langle v_i, R_{v_d}(x) \rangle > 0\}$, $i = 1, \ldots, d-1$, and reason about the value $\langle v_d, x \rangle$ takes in each.

First, we define the conditioning set $E := \{x \colon \langle v_i, x \rangle > 0, \forall i = 1, \ldots, d-1\}$ and observe that $\mathbb{P}(X \in E) > 0$ so all we need to prove is $\mathbb{E}[\mathbb{1}_{\langle v_d, X \rangle > 0}\mathbb{1}_E] > \mathbb{E}[\mathbb{1}_{\langle v_d, X \rangle \leq 0}\mathbb{1}_E]$, where $\mathbb{1}_E$ is the indicator function of the set $E$. To do so, we define $U := \{x \colon \langle v_i, R_{v_d}(x) \rangle > 0, \forall i = 1, \ldots, d-1\}$, $A_+ := E \cap U$, $A_- := E \cap U^c$, split the integral $\int_E \mathbb{1}_{\{\langle v_d, X \rangle > 0\}}(x)\phi(x)\,\mathrm{d}x$ into $\int_{A_+} \mathbb{1}_{\{\langle v_d, X \rangle > 0\}}(x)\phi(x)\,\mathrm{d}x + \int_{A_-} \mathbb{1}_{\{\langle v_d, X \rangle > 0\}}(x)\phi(x)\,\mathrm{d}x$ ($\phi$ is the standard normal density function; analogously for $\mathbb{1}_{\{\langle v_d, X \rangle \leq 0\}}$), and consider $X \in A_+$ and $X \in A_-$ separately:

(I) $X \in A_+$: Take any $x, v \in \mathbb{R}^d$ and define the orthogonal projection map on $v$, $B_v := vv^\top/\|v\|_2^2$, and the corresponding projections of $x$, $x_v := B_v x$, $x_v^\perp = (I - B_v)x$, so that $x = x_v + x_v^\perp$. Since

$$\|x\|_2^2 = \|x_v + x_v^\perp\|_2^2 = \|x_v\|_2^2 + \|x_v^\perp\|_2^2 = \|-x_v + x_v^\perp\|_2^2 = \|R_v(x)\|_2^2 \,,$$

it follows that $\phi(x) = \phi(R_{v_d}(x))$. Noticing further that $R_{v_d}(x) = (I - 2B_{v_d})x$ and recalling $R_{v_d}(R_{v_d}(x)) = x$, we have $|\det \nabla_x R_{v_d}(x)| = |-1| = 1$. The crucial observation here is $\langle x, v_d \rangle > 0 \iff \langle x_{v_d}, v_d \rangle > 0$, $\langle x, v_d \rangle \leq 0 \iff \langle R_{v_d}(x), v_d \rangle > 0$ (up to null sets), and that $A_+ = R_{v_d}[A_+] = \{R_{v_d}(x) \colon x \in A_+\}$ which follows from the definition of the set $A_+$. In particular this means that whenever $x \in A_+$ then also $-x \in A_+$, and thus by the above established symmetry and the change of variable formula, $\int_{A_+} \mathbb{1}_{\{\langle v_d, X \rangle > 0\}}(x)\phi(x)\,\mathrm{d}x = \int_{A_+} \mathbb{1}_{\{\langle v_d, X \rangle \leq 0\}}(x)\phi(x)\,\mathrm{d}x$, i.e. the conditional probabilities of both $A_+ \cap \{\langle v_d, X \rangle > 0\}$ and $A_+ \cap \{\langle v_d, X \rangle \leq 0\}$ are equal.

(II) $X \in A_-$: Notice that for any $i = 1, \ldots, d-1$

$$\langle v_i, R_{v_d}(x) \rangle = \langle v_i, x \rangle - 2\frac{\langle v_d, x \rangle}{\|v_d\|_2^2}\langle v_d, v_i \rangle \,.$$

Hence if $\langle v_d, x \rangle \leq 0$ then $\langle v_i, R_{v_d}(x) \rangle \geq \langle v_i, x \rangle > 0$ from the definition $\langle v_d, v_i \rangle = \mathrm{Cov}(\Delta_d, \Delta_i)$ and the assumption $\mathrm{Cov}(\Delta_d, \Delta_i) > 0$. Now by the definition of $U$ in $A_- = E \cap U^c$, for any $x \in A_-$, there must exist $i \in \{1, \ldots, d-1\}$ such that $\langle v_i, R_{v_d}(x) \rangle \leq 0$ which implies $\langle v_d, x \rangle > 0$ by the above argument. It is thus sufficient to establish $\mathbb{P}(X \in A_-) > 0$ to complete the proof as the intersection $A_- \cap \{\langle v_d, X \rangle \leq 0\}$ is empty.

Since $\langle v_d, v_i \rangle = \mathrm{Cov}(\Delta_d, \Delta_i) > 0$, $v_d \in E$ and $\langle v_i, R_{v_d}(v_d) \rangle = -\langle v_i, v_d \rangle < 0$, $\forall i = 1, \ldots, d-1$, we have $v_d \in A_-$. We can thus construct a convex polytope $V \subseteq A_-$ such that $\mathbb{P}(X \in V) > 0$. Specifically, pick some $i \in \{1, \ldots, d-1\}$, for example $i = \arg\max_{i \in \{1,\ldots,d-1\}} \langle v_d, v_i \rangle$, and set $\kappa := \max_{k,l \in \{1,\ldots,d\}} |\langle v_k, v_l \rangle| = \max_{k \in \{1,\ldots,d\}} \|v_k\|_2^2 > 0$. Now define

$$V := \left\{x \colon x = u + v_d + \sum_{j=1}^{d-1} \alpha_j v_j \,, \alpha_j \in [0, \tfrac{\langle v_d, v_i \rangle}{\kappa(d-1)}) \,, u \in \mathrm{span}(v_1, \ldots, v_d)^\perp\right\},$$

where $\mathrm{span}(v_1, \ldots, v_d)^\perp$ is the orthogonal complement of the linear span of the vectors $(v_1, \ldots, v_d)$. Clearly $V \subseteq E$ as for any $x \in V$, $\langle v_i, x \rangle > 0$ from the bound on the coefficients $\alpha$. To see that $x \in V \implies x \in U^c$, note

$$\langle v_i, R_{v_d}(x) \rangle = -\langle v_i, v_d \rangle + \sum_{j=1}^{d-1} \underbrace{\alpha_j}_{\geq 0}\Big[\langle v_i, v_j \rangle - 2\underbrace{\frac{\langle v_d, v_i \rangle}{\|v_d\|_2^2}\langle v_j, v_d \rangle}_{> 0}\Big].$$

Since the first and last terms are strictly negative, we just need to control the second term. We again apply the definition of $V$ to bound $\sum_j \alpha_j \langle v_i, v_j \rangle < \langle v_i, v_d \rangle$ which implies $\langle v_i, R_{v_d}(x) \rangle < 0$ for every $x \in V$. Thus $V \subseteq A_-$ and because $V$ has non-zero volume, its probability under $\mathcal{N}(0, I)$ will be positive. Hence $\int_{A_-} \mathbb{1}_{\{\langle v_d, X \rangle > 0\}}(x)\phi(x)\,\mathrm{d}x > \int_{A_-} \mathbb{1}_{\{\langle v_d, X \rangle \leq 0\}}\phi(x)\,\mathrm{d}x = 0$. $\qquad \square$

We are now ready to prove lemma 4.

**Lemma 4** (Formal statement). *Let $\hat{\pi} \sim P_{\hat{\pi}} = G_{\#}P_{\hat{Q}^{\pi}}$ where $\hat{Q} \sim P_{\hat{Q}^{\pi}}$ is the SU model for the uniform policy $\pi$. For $2 \leq k < 2L$ even, define $U_k = \{\hat{\pi} : \hat{\pi}(s_0) = \ldots = \hat{\pi}(s_{k-2}) = \delta_{\mathrm{UP}}\}$ where $\delta_{\mathrm{UP}}$ is the policy of selecting $\mathrm{UP}$ with probability one. Then $P_{\hat{\pi}}(\hat{\pi}(s_k) = \delta_{\mathrm{UP}} \mid \hat{\pi} \in U_k) > 1/2$ if there exists an even $0 \leq j < k$ such that $\mathrm{Cov}(\hat{Q}(s_k, \mathrm{UP}), \hat{Q}(s_j, \mathrm{UP})) > \mathrm{Cov}(\hat{Q}(s_k, \mathrm{DOWN}), \hat{Q}(s_j, \mathrm{UP}))$.*

*Proof.* Under $P_{\hat{\pi}}$, $G(\hat{Q}) = \delta_{\mathrm{UP}}$ iff $\Delta_{k/2} = \hat{Q}(s_k, \mathrm{UP}) - \hat{Q}(s_k, \mathrm{DOWN}) > 0$. By lemma 7, the distribution of the random vector $\Delta = [\Delta_0, \Delta_1, \ldots, \Delta_{k/2}]^{\top}$ is a zero mean Gaussian, and in particular

$$P_{\hat{\pi}}(\hat{\pi} = \delta_{\mathrm{UP}} \mid \hat{\pi} \in U_k) = \mathbb{P}(\Delta_{k/2} > 0 \mid \Delta_0 > 0, \ldots, \Delta_{k/2-1} > 0).$$

To prove the desired claim, we therefore need to show that existence of even $0 \leq j < k$ such that $\mathrm{Cov}(\hat{Q}(s_k, \mathrm{UP}), \hat{Q}(s_j, \mathrm{UP})) > \mathrm{Cov}(\hat{Q}(s_k, \mathrm{DOWN}), \hat{Q}(s_j, \mathrm{UP}))$, implies $\mathbb{P}(\Delta_{k/2} > 0 \mid \Delta_0 > 0, \ldots, \Delta_{k/2-1} > 0) > 1/2$. The statement follows from:

$$\mathrm{Cov}(\hat{Q}(s_k, \mathrm{UP}), \hat{Q}(s_j, \mathrm{UP})) > \mathrm{Cov}(\hat{Q}(s_k, \mathrm{DOWN}), \hat{Q}(s_j, \mathrm{UP})), \text{ for some even } 0 \leq j < k$$

$$\overset{\text{lemma 8}}{\Longleftrightarrow} \mathrm{Cov}(\hat{Q}(s_k, \mathrm{UP}), \hat{Q}(s_j, \mathrm{UP})) > \mathrm{Cov}(\hat{Q}(s_k, \mathrm{DOWN}), \hat{Q}(s_j, \mathrm{UP})), \text{ for all even } 0 \leq j < k$$

$$\overset{\text{lemma 7}}{\Longleftrightarrow} \mathrm{Cov}(\Delta_{k/2}, \Delta_{j/2}) > 0, \text{ for all even } 0 \leq j < k$$

$$\overset{\text{lemma 9}}{\Longleftrightarrow} \mathbb{P}(\Delta_{k/2} > 0 \mid \Delta_0 > 0, \ldots, \Delta_{k/2-1} > 0) > 1/2.$$

$\square$

**Proposition 3** (Formal statement). *Assume the SU model with: (i) one-hot state-action embeddings $\phi$, (ii) uniform exploration thus far, (iii) successor representations learnt to convergence for a uniform policy. For $2 \leq k < 2L$ even, let $s_k$ be a state visited $N$ times thus far, and $\pi$, $\hat{Q} \sim P_{\hat{Q}^{\pi}}$, $\hat{\pi} \sim P_{\hat{\pi}}$, and $U_k$ be defined as in lemma 4. Then*

$$P_{\hat{\pi}}(\hat{\pi}(s_k) = \delta_{\mathrm{UP}} \mid \hat{\pi} \in U_k) > P_{\hat{\pi}}(\hat{\pi}(s_k) = \delta_{\mathrm{DOWN}} \mid \hat{\pi} \in U_k),$$

*with probability greater than $1 - \epsilon_N$, where $\epsilon_N < 0.75^N e^{-\frac{N}{50}} + (1 - 0.75^N)e^{-0.175N}$.*

*Proof.* By lemma 4, we know that $P_{\hat{\pi}}(\hat{\pi}(s_k) = \delta_{\mathrm{UP}} \mid \hat{\pi} \in U_k) > P_{\hat{\pi}}(\hat{\pi}(s_k) = \delta_{\mathrm{DOWN}} \mid \hat{\pi} \in U_k)$ holds if $\mathrm{Cov}(\hat{Q}(s_k, \mathrm{UP}), \hat{Q}(s_j, \mathrm{UP})) > \mathrm{Cov}(\hat{Q}(s_k, \mathrm{DOWN}), \hat{Q}(s_j, \mathrm{UP}))$ for some $j = 0, 2, \ldots, k - 2$. By lemma 8, this condition is equivalent to requiring $\mathbb{V}(\hat{Q}(s_k, \mathrm{UP})) > \mathbb{V}(\hat{Q}(s_k, \mathrm{DOWN}))$. Our approach is thus based on lower bounding the probability of the event

$$\{\hat{Q} : \mathbb{V}(\hat{Q}(s_k, \mathrm{UP})) > \mathbb{V}(\hat{Q}(s_k, \mathrm{DOWN}))\}. \tag{5}$$

The rest of the proof is divided into two stages:

(I) We derive a crude bound $\Upsilon_1(\hat{Q}(s_k, \mathrm{UP})) \leq \mathbb{V}(\hat{Q}(s_k, \mathrm{UP}))$ and compute a lower bound on the probability of the event $\Upsilon_1(\hat{Q}(s_k, \mathrm{UP})) > \mathbb{V}(\hat{Q}(s_k, \mathrm{DOWN}))$.

(II) We then derive a tighter lower bound $\Upsilon_2(\hat{Q}(s_k, \mathrm{UP}))$, and again compute a lower bound on the probability of the event $\Upsilon_2(\hat{Q}(s_k, \mathrm{DOWN})) > \mathbb{V}(\hat{Q}(s_k, \mathrm{DOWN}))$.

(I) The bound $\Upsilon_1(\hat{Q}(s_k, \mathrm{UP})) \leq \mathbb{V}(\hat{Q}(s_k, \mathrm{UP}))$ will correspond to a worst case assumption about the distribution of data available from exploration, and $\Upsilon_2(\hat{Q}(s_k, \mathrm{UP}))$ to a less pessimistic scenario. The change of setup involved in moving from the first bound to the second will be illustrative of the manner in which, under the SU model, the more states the agent has previously observed beyond $s_k$, the more likely it is to satisfy the condition from equation (5) and consequently $\mathrm{Cov}(\hat{Q}(s_k, \mathrm{UP}), \hat{Q}(s_j, \mathrm{UP})) > \mathrm{Cov}(\hat{Q}(s_k, \mathrm{DOWN}), \hat{Q}(s_j, \mathrm{UP}))$ for all $j = 0, 2, \ldots, k - 2$.

From lemma 6, we know that the SU model of rewards will be a zero mean Gaussian with a diagonal covariance. In particular, the covariance takes the form $(\theta^{-1}I + \beta^{-1}\sum_t \phi_t \phi_t^{\top})^{-1}$, where recall $\theta$ is the prior and $\beta$ is the likelihood variance, implying that the diagonal entries will be $\nu(n) := (\theta^{-1} + \beta^{-1}n)^{-1}$ where $n$ is the number of times the corresponding state-action was observed.

Recall that the agent has previously visited the state $s_k$ $N$ times. We will write $N_1$ for the number of times we have observed $(s_k, \text{UP})$ so far, $N_2$ for the number of times $(s_k, \text{UP})$ *and* $(s_{k+2}, \text{UP})$ have both been observed within a single episode, and so forth. Observe

$$\mathbb{V}(\hat{Q}(s_k, \text{UP})) = \nu(N_1) + 2^{-1}(\nu(N_2) + \nu(N_1 - N_2)) +$$
$$\mathbb{1}_{N_3 > 0} 2^{-2}(\nu(N_3) + \nu(N_2 - N_3) + \mathbb{1}_{N_4 > 0} \ldots)$$
$$\geq \nu(N_1) + 2^{-1}(\nu(N_2) + \nu(N_1 - N_2))$$

We now minimise $\nu(N_2) + \nu(N_1 - N_2)$ with respect to $N_2$, finding the minima to occur at $N_2 = N_1$ and $N_2 = 0$, in both cases giving the bound

$$\Upsilon_1(\hat{Q}(s_k, \text{UP})) := \frac{3}{2}\nu(N_1) + \frac{1}{2}\nu(0) \leq \mathbb{V}(\hat{Q}(s_k, \text{UP}))$$

This bound can be interpreted as assuming that after taking action UP, the agent has always proceeded to move DOWN, thus terminating the episode. We now compute a lower bound on the probability that $\Upsilon_1(\hat{Q}(s_k, \text{UP})) > \mathbb{V}(\hat{Q}(s_k, \text{DOWN}))$, in terms of $N_1$. We have

$$\Upsilon_1(\hat{Q}(s_k, \text{UP})) - \mathbb{V}(\hat{Q}(s_k, \text{DOWN})) = \frac{3}{2}\nu(N_1) - \nu(N - N_1) + \frac{1}{2}\nu(0) > \frac{3}{2}\nu(N_1) - \nu(N - N_1)$$

which is greater than zero when $\theta^{-1} + \beta^{-1}(3N - 5N_1) > \beta^{-1}(3N - 5N_1) > 0$, i.e. whenever $N_1 < \frac{3N}{5}$. By Hoeffding's inequality, $\mathbb{P}(N_1 \geq \frac{(1+\delta)N}{2}) \leq e^{-\frac{\delta^2 N}{2}}$. Thus, letting $\delta = 5^{-1}$, $\mathbb{V}(\hat{Q}(s_k, \text{UP})) > \mathbb{V}(\hat{Q}(s_k, \text{DOWN}))$ holds with probability greater than $1 - e^{-\frac{N}{50}}$.

(II) Notice that we have obtained the $\Upsilon_1$ bound by considering the worst case scenario for $N_2$, namely $N_2 = 0$. Here we derive a tighter bound by treating the two cases, $N_2 = 0$ and $N_2 > 0$, separately. For $N_2 > 0$, we follow an approach analogous to (I): we assume the "next" worst-case scenario, which is easily seen to be $N_3 = 0$, and compute a lower bound on $\mathbb{V}(\hat{Q}(s_k, \text{UP}))$

$$\Upsilon_2(\hat{Q}(s_k, \text{UP})) := \nu(N_1) + \nu(N_2) + \frac{1}{2}\nu(N_1 - N_2).$$

After some algebra, we obtain $\Upsilon_2(\hat{Q}(s_k, \text{UP})) > \mathbb{V}(\hat{Q}(s_k, \text{DOWN}))$ for all $N_2 > 0$ and $N_1 \leq \frac{1}{41}(27 + 4\sqrt{2})N =: c$. We thus only need to bound the probability of $N_1 > c$. Using Hoeffding's inequality as in (I) for a suitably chosen $\delta$, we see $\mathbb{P}(N_1 > c) \leq \exp\{-\frac{(13+8\sqrt{2})^2}{3362}N\} < e^{-0.175N}$. For $N_2 = 0$, we use the bound from part (I), and thus the only thing remaining is to compute the probability of $N_2 = 0$:

$$\mathbb{P}(N_2 = 0) = \sum_{K=0}^{N}\mathbb{P}(N_2 = 0 \mid N_1 = K)\mathbb{P}(N_1 = K) = \sum_{K=0}^{N} 2^{-K} 2^{-N}\binom{N}{K}$$
$$= \sum_{K=0}^{N}\binom{N}{K} 4^{-K} 2^{K-N} = (4^{-1} + 2^{-1})^N = 0.75^N.$$

Combining the above results, we see that $\mathbb{V}(\hat{Q}(s_k, \text{UP})) > \mathbb{V}(\hat{Q}(s_k, \text{DOWN}))$ will hold with probability greater than $1 - \epsilon_N$ where $\epsilon_N < 0.75^N e^{-\frac{N}{50}} + (1 - 0.75^N)e^{-0.175N}$. $\qquad\square$

## B.2 Proofs for section 5.3

The following is an extension of proposition 5 to activations such as ReLU, Leaky ReLU, or Tanh.

**Proposition 10.** *Consider the same setting as in proposition 5 with the exception that $\varphi$ for which $\varphi[(0, \infty)] = \{\varphi(x) : x > 0\} \subseteq (0, \infty)$. Then sampling independently form the prior $w_a \sim \mathcal{N}(0, \sigma_w^2 I)$, $U_{hs} \sim \mathcal{N}(0, \sigma_u^2)$ solves a tied action binary tree of size $L$ in $T \leq -[\log_2(1 - 2^{-d}(1 - 2^{-d})^L)]^{-1}$ median number of episodes, or approximately $-[\log_2(1 - 2^{-d})]^{-1}$ for $d \geq 10$.*

*Proof.* As in the proof of proposition 5, let us define $\Delta := w_{\text{UP}} - w_{\text{DOWN}}$ and observe UP is selected if $\hat{Q}(s, \text{UP}) - \hat{Q}(s, \text{DOWN}) = \langle\phi(s), w_{\text{UP}} - w_{\text{DOWN}}\rangle > 0$. We can thus lower bound

$$\mathbb{P}\Big[\bigcap_{j=0}^{L-1}\{\hat{Q}(s_{2j}, \text{UP}) > \hat{Q}(s_{2j}, \text{DOWN})\}\Big] \geq \mathbb{P}\Big[\bigcap_{j=0}^{L-1}\{\langle\phi(s_{2j}), \Delta\rangle > 0\} \mid \Delta > 0\Big]\mathbb{P}(\Delta > 0),$$

where $\Delta > 0$ is meant elementwise. As $\Delta \sim \mathcal{N}(0, 2\sigma_w^2 I)$, $\mathbb{P}(\Delta > 0) = 2^{-d}$ for all $L$. By independence $\mathbb{P}[\bigcap_{j=0}^{L-1}\{\langle\phi(s_{2j}), \Delta\rangle > 0\} \mid \Delta > 0] = \prod_{j=0}^{L-1}\mathbb{P}(\{\phi(s_{2j}) > 0\})$ where $>$ is to be

interpreted elementwise. From the assumption $\varphi[(0, \infty)] \subseteq (0, \infty)$ and the assumed $\phi(s) = \varphi(U1_s)$, $U_{hs} \sim \mathcal{N}(0, \sigma_u^2)$, we have $\mathbb{P}(\{\phi(s) > 0\}) \geq 1 - 2^{-d}$, which implies that probability of success within a single episode is lower bounded by $2^{-d}(1 - 2^{-d})^L$. The result follows by substituting this probability into the formula for the median of a geometric distribution. $\square$

## C   Appendix to section 5: implementation & experimental details

Pseudocode for SU. Quantities superscripted with $\dagger$ are treated as fixed during optimisation.

---

**Algorithm 1** Successor Uncertainties with posterior sampling

---

**Require:** Neural networks $\hat{\psi}$ and $\hat{\phi}$; weight vector $\hat{w}$; prior variance $\theta > 0$; likelihood variance $\beta > 0$; covariance decay factor $\zeta \in [0, 1]$; BATCH_SIZE $\in \mathbb{N}$; LEARNING_RATE $> 0$; environment ENV; action set $\mathcal{A}$; discount factor $\gamma \in [0, 1)$.

   initialise $\Lambda \leftarrow \theta^{-1}I$, $\hat{\Sigma}_w \leftarrow \Lambda^{-1}$
   **for each** episode **do**
      sample $w \sim N(\hat{w}, \hat{\Sigma}_w)$
      $s \leftarrow$ ENV.RESET()

      **repeat**
         $a \leftarrow \text{argmax}_{z \in \mathcal{A}} \langle \hat{\psi}(s, z), w \rangle$
         $s', r, done \leftarrow$ ENV.INTERACT$(s)$
         $\mathcal{D} \leftarrow \mathcal{D} \cup \{(s, a, r, s', done)\}$

         $\mathcal{B} \sim$ UNIFORM$(\mathcal{D}, \text{BATCH\_SIZE})$
         $\ell \leftarrow \sum_{b \in \mathcal{B}}$ SU_LOSS$(b, \hat{\Sigma}_w)$
         $\hat{\phi}, \hat{\psi}, \hat{w} \leftarrow$ SGD.STEP$(\ell, \text{LEARNING\_RATE})$

         $\Lambda \leftarrow \zeta \Lambda + \beta^{-1} \hat{\phi}(s, a) \hat{\phi}(s, a)^\top$
         $s \leftarrow s'$
      **until** $done$

      $\hat{\Sigma}_w \leftarrow \Lambda^{-1}$
   **end for**

   **function** SU_LOSS(EXPERIENCE_TUPLE, $\hat{\Sigma}_w$)
      $s, a, r, s, done \leftarrow$ EXPERIENCE_TUPLE
      sample $w \sim N(\hat{w}, \hat{\Sigma}_w)$
      $a' \leftarrow \text{argmax}_{z \in \mathcal{A}} \langle \hat{\psi}(s, z), w \rangle$

$$y_Q \leftarrow \begin{cases} 0 & \text{if } done \\ \gamma \langle \hat{w}, \hat{\psi}(s', a') \rangle & \text{otherwise} \end{cases}$$

$$y_{SF} \leftarrow \begin{cases} 0 & \text{if } done \\ \gamma \hat{\psi}(s', a') & \text{otherwise} \end{cases}$$

      **return** $|\langle \hat{w}, \hat{\phi}(s, a) \rangle - r|^2 + \|\hat{\psi}(s, a) - \hat{\phi}(s, a) - y_{SF}^\dagger\|_2^2 + |\langle \hat{w}, \hat{\psi}(s, a) \rangle - r - y_Q^\dagger|^2$
   **end function**

---

### C.1   Appendix to sections 5.1 and 5.2: tabular experiments

**Neural network architecture**   The architecture used for tabular experiments consists of:

1.  A neural network mapping one-hot encoded state vectors and one-hot encoded action vectors to a hidden layer $\hat{\phi}(s, a)$, and then to reward prediction $\hat{r}(s, a)$ via weights $\hat{w}$. Weights mapping state vectors to hidden layer are initialised using a folded Xavier normal initialisation and followed by ReLU activation. Weights $\hat{w}$ are initialised to zero, consistent with a Bayesian linear regression model with a zero mean prior.

2. A set of weights that linearly maps state-action vectors to $\hat{\psi}(s, a)$.

**Binary tree MDP** Table 3 contains the hyperparameters considered during gridsearch and the final values used to produce figure 2. Hyperparameter values are not included for UBE and BDQN, as they do not affect performance (that is, BDQN and UBE perform uniformly random exploration for all hyperparameter settings). All methods used one layer fully connected ReLU networks, Xavier initialisation, and a replay buffer of size 10,000. Hyperparameters for all methods were selected by gridsearch on a $L = 100$ sized binary tree. Hyperparameters were then kept fixed as binary tree size $L$ was varied.

Table 3: Binary tree experiment algorithm hyperparameters gridsearch sets and values used for Successor Uncertainties, Bootstrap+Prior (1x compute) and Bootstrap+Prior (25x compute).

| Hyperparameter | Gridsearch set | Algorithm | | |
|---|---|---|---|---|
| | | SU | B+P 1x | B+P 25x |
| Gradient steps per episode | — | 10 | 10 | 250 |
| Hidden size | $\{20, 40\}$ | 20 | 20 | 20 |
| Prior variance $\theta$ | $\{1, 10^2, 10^4\}$ | $10^4$ | — | — |
| Likelihood variance $\beta$ | $\{10^{-3}, 10^{-2}, 10^{-1}\}$ | $10^{-3}$ | — | — |
| $\hat{\Sigma}_w$ decay factor $\zeta$ | — | 1 | — | — |
| Ensemble size $K$ | $\{10, 20, 40\}$ | — | 10 | 10 |
| Bootstrap probability | $\{0.1, 0.25, 0.75, 0.9, 1.0\}$ | — | 0.75 | 1.0 |
| Prior weight | $\{0.0, 0.1, 1.0, 10.0\}$ | — | 0.1 | 0.0 |

**Chain MDP** Problem description copied verbatim from Osband et al. (2018):

> *The environments are indexed by problem scale $L \in \mathbb{N}$ and action mask $W \sim \mathrm{Ber}(0.5)^{L \times L}$, with $\mathcal{S} = \{0, 1\}^{L \times L}$ and $\mathcal{A} = \{0, 1\}$. The agent begins each episode in the upper left-most state in the grid and deterministically falls one row per time step. The state encodes the agent's row and column as a one-hot vector $s_t \in \mathcal{S}$. The actions $\{0, 1\}$ move the agent left or right depending on the action mask $W$ at state $s_t$, which remains fixed. The agent incurs a cost of $0.01/L$ for moving right in all states except for the right-most, in which the reward is $1$. The reward for action left is always zero. An episode ends after $L$ time steps so that the optimal policy is to move right each step and receive a total return of $0.99$; all other policies receive zero or negative return.*

Table 4 contains the hyperparameter settings used to produce the results in figure 3. We were unable to run experiments with $L > 160$ for Successor Uncertainties due to memory limitations. $|\mathcal{S}|$ scales as $\mathcal{O}(L^2)$ for this problem. Consequently, with one hot encoding, the required neural network weight vectors required grew too large. A smarter implementation using a library designed for operating on sparse embeddings would alleviate this problem.

Table 4: Hyperparameters used for Successor Uncertainties in chain experiments. Hidden size fixed at 20 to match architecture in Osband et al. (2018).

| Hyperparameter | Gridsearch set | Value used |
|---|---|---|
| Gradient steps per episode | $\{10, 20, 40\}$ | 40 |
| Hidden size | — | 20 |
| Prior variance $\theta$ | $\{1, 10^2, 10^4\}$ | 1 |
| Likelihood variance $\beta$ | $\{10^{-3}, 10^{-2}, 10^{-1}\}$ | $10^{-2}$ |
| $\hat{\Sigma}_w$ decay factor $\zeta$ | — | 1 |

## C.2 Appendix to section 6: Atari 2600 experiments

**Training procedure**  We train for 200M frames (50M action selections with each action repeated for 4 frames), using the ADAM optimiser (Kingma & Ba, 2014) with a learning rate of $5 \times 10^{-5}$ and a batch size of 32. A target network is utilised, as in Mnih et al. (2015), and is updated every $10,000$ steps, as in Van Hasselt et al. (2016).

**Network architecture**  We use a single neural network to obtain estimates $\hat{\phi}$ and $\hat{\psi}$.

1. Features: the neural network converts $4 \times 84 \times 84$ pixel states (obtained through standard frame max-pooling and stacking) into a 3136-dimensional feature vector, using a convolution network with the same architecture as in Mnih et al. (2015).

2. Hidden layer: the feature vector is then mapped to a hidden representation of size 1024 by a fully connected layer followed by a ReLU activation.

3. $\hat{\phi}$ prediction: the hidden representation is mapped to a size 64 prediction of $\hat{\phi}$ for each action in $\mathcal{A}$ by a fully connected layer with ReLU activation.

4. $\hat{\psi}$ prediction: the hidden representation is mapped to $1 + |\mathcal{A}|$ vectors of size 64. The first vector gives the average successor features for that state $\bar{\psi}(s)$, whilst each of the $|\mathcal{A}|$ vectors predicts an advantage $\tilde{\psi}(s,a)$. The overall successor feature prediction is given by $\hat{\psi}(s,a) = \bar{\psi}(s) + \tilde{\psi}(s,a)$.

5. Linear $\hat{Q}^\pi$ and $\hat{r}$ prediction: a final linear layer with weights $\hat{w}$ maps $\hat{\phi}$ to reward prediction and $\hat{\psi}$ to Q value prediction with both predictors sharing weights.

**Hyperparameter selection**  We used six games for hyperparameter selection: ASTERIX, ENDURO, FREEWAY, HERO, QBERT, SEAQUEST, a subset of the games commonly used for this purpose (Munos et al., 2016). 12 combinations of parameters in the 'search set' column were tested (that is, not an exhaustive gridsearch), for a total of $12 \times 6 = 72$ full game runs, or approximately 33% of the entire computational cost of the experiment.

Table 5: Hyperparameters used for Successor Uncertainties in Atari 2600 experiments.

| Hyperparameter | Search set | Value used |
|---|---|---|
| Action repeat | — | 4 |
| Train interval | — | 4 |
| Learning rate | $\{2.5 \times 10^{-4}, 5 \times 10^{-5}\}$ | $5 \times 10^{-5}$ |
| Batch size | — | 32 |
| Gradient clip norm cutoff | — | 10 |
| Target update interval | $\{10^3, 10^4\}$ | $10^4$ |
| Successor feature size | $\{32, 64\}$ | 64 |
| Hidden layer size | — | 1024 |
| Prior variance $\theta$ | — | 1 |
| Likelihood variance $\beta$ | $\{10^{-3}, 10^{-2}\}$ | $10^{-3}$ |
| $\hat{\Sigma}_w$ decay factor $\zeta$ | $\{1 - 10^{-5}, 1 - 10^{-4}\}$ | $1 - 10^{-5}$ |