[Reviews · NeurIPS 2019]

Reviewer 1



Originality: the paper proposes to address exploration in reinforcement learning using a combination of posterior sampling, successor features, and Bayesian linear regression. To the best of my knowledge the proposed combination is novel. The authors also do a good job contextualizing the proposed method within the related literature. Quality: the paper is well written and seems to be technically correct. The claims are supported by both theoretical and empirical results. The authors are also upfront about the limitations of the proposed approach (Section 4.4). Clarity: although the paper is well written, the presentation could perhaps be slightly improved in two points. First, with the exception of Fig. 1, there is a lack of intuitive explanations that may make it difficult for a reader less familiar with the subject to grasp the ideas at first. Second, the narrative behind SU seems a bit entangled with the use of neural networks --e.g., line 19--, although some of the theoretical arguments in favor of it actually rely on a tabular representation. I wonder if it is possible to present the SU concept in isolation and later argue that it has as one of its benefits the fact that it can be easily combined with complex function approximators. Significance: the paper proposes a method to perform "deep exploration" in RL. The method is simple and has low computational cost --as discussed in line 188, it can be seen as a small modification of previous methods resulting from the structure imposed in (5). As such, it seems to me that it has the potential of being adopted by the community and also serving as an inspiration for future research. Post-rebuttal update: ------ As stated in my rebuttal, I think this paper is quite strong content-wise. It could benefit from a clearer presentation, though, as indicated in the reviews. I hope the authors put some effort in making the next version of the paper more accessible.

Reviewer 2



This paper proposes using Bayesian linear regression to get a posterior over successor features as a way of representing uncertainty, from which they sample for exploration. I found the characterization of Randomised Policy Iteration to be strange, as it only seems to apply to UBE but not bootstrapped DQN, With bootstrapped DQN, each model in the ensemble is a value function pertaining to a different policy, thus there is no single reference policy. The ensemble is trying to represent a distribution of optimal value functions, rather than value functions for a single reference policy. Proposition 1: In the case of neural networks, and function approximation in general, it is very unlikely that we will get a factored distribution, so this claim does not seem applicable in general. In fact, in general there should be very high correlation between the q-values between nearby states. Is this claim a direct response to UBE? Also the analysis fixes the policy to consider the distribution of value functions, but this seems to not be how posterior sampling is normally considered, but rather only the way UBE considers it. A straightforward approach to posterior sampling would consider a distribution over optimal value functions, rather than being tied to any specific policy. It is confusing that this analysis is presented as being standard in posterior sampling literature. Proposition 2: While yes, you can easily change your value function to maintain the same policy, and thereby breaking any constraint that your value function may satisfy, this does not mean that it is a good idea to get rid of propagation of uncertainty. The key issue to consider is what happens as your posterior changes. Is there a consistent way to update your distribution of value functions that keeps it matching the posterior sampling policy as the posterior is updated with new data? This proposition only holds for a fixed policy, but PSRL is not sampling from the posterior of models according to a fixed policy. So even if the sampling policies match, they only partially match the true sampling policy of PSRL, which seems quite limited. This limitation is mentioned later on in the paper, but it is unclear why proposition 2 is a desirable quality to have in the first place. Wouldn't one want to match the true sampling policy of PSRL, even at the cost of violating proposition 2? Also the proposed Successor Uncertainties algorithm still propagates uncertainties, so it's even more unclear what the purpose of this proposition is. The experimental results in chain and tower MDPs are quite promising for SU. It seems to show that SUs gracefully scale down to tabular settings. The results in Atari do seem to show improvement over bootstrapped DQN. Overall, the idea of using Bayesian linear regression on successor features seems to be a promising direction for research, and the experimental results back this up. However the theoretical analyses are confusing and not well motivated. **Update after rebuttal** Thanks to the authors for willing to clarify the definitions and propositions. With more clarity and motivation, I am willing to move to a 7.

Reviewer 3



The work starts by highlighting potential limitations of PSRL-inspired model-free exploration methods with intuitive propositions. To overcome these drawbacks a framework for decoupling the reward and transition uncertainties by modelling the reward function via BLR is proposed. This architecture is used along with a TD-successor feature learning with an additional Q-values constraint. They conclude with experiments on tabular MDPs and Atari 2600 games. Originality - The decoupling of uncertainty via BLR of rewards is an interesting direction for driving exploration. Quality - The first-half of the paper is very clearly written. But the second half from Section 4 is lacking adequate motivation for the method and structure. Clarity - The part from Section 4 while a proposal for the algorithm, lacks adequate motivation for why the approach would not suffer from the drawbacks highlighted in part 1. It is unclear whether SU is an interesting exploration algorithm because it overcomes the limitation of Proposition 2, or because it satisfies Definition 2. While the authors acknowledge some limitations of the method in Section 4.4, the two parts of the paper seem rather incongruent. Further (1) algorithmically it is unclearn how the uncertainties drive exploration — greedily/stochastically? (2) there is a disconnect between the pseudocode in the appendix and the last paragraph of 4.2 (3) while the complete theoretical section in Section 5.2 is wrt Figure 1 is interesting, the failure of BDQN and UBE is surprising — if the constants are high, I do not see how they fail so much. (4) Section 5.3 seems unnecessary and is rather unclearly presented (5) y-axis in Figure 4 — clipped or is-between? (6) Why do the embedding need to satisfy said properties in Section 4.1? (7) Successor Uncertainties seems to be a confusing name considering the proposal models only the uncertainty in the reward function explicitly. How does modelling reward uncertainty compare to modelling transition uncertainty? I understand it is briefly discussed in Section 4.4., but the discussion seems to say "we can benefit from modelling successor uncertainty" - mostly rendering the name a misnomer. (8) Section 5.2 is rather unclear — tied actions == stochastic transitions? (9) the neural network models the action as just another input? Significance - I think the work is significant in parts but the complete paper can be better organized, and the contributions of Part 2 better placed in the context of Part 1. While modelling the reward uncertainty seems promising for prorogation of uncertainty in a more robust manner, the presentation of the actual algorithm obfuscates a lot of details in the main paper. PS: I have not reviewed proofs for Section 5.1. ------ Post-rebuttal: Thank you for your clarifying remarks, and sorry about the confusion regarding the Chain MDP. I have read the rebuttal and have updated my score.

[Author Response · NeurIPS 2019]

We thank the reviewers for their valuable feedback. A common point raised by the reviewers is that the clarity of the
paper could be improved. We agree and, as suggested by the reviewers, will move some proofs to the appendix and
replace them with more intuitive explanations. We address reviewer specific comments below:

**R3**: We thank R3 for recognising the value of decoupling reward and transition uncertainty, and of the intuition provided
by our theory about the failure modes of existing methods. R3 says: "[SU] lacks adequate motivation for why the
approach would not suffer from the drawbacks highlighted in part 1". SU is motivated in Section 3 (please see lines
81-85 and the ensuing discussion). We satisfy our Definition 2 which allows us to avoid the pitfalls highlighted in
Section 3 (please see lines 146-148). The mechanism through which SU avoids these issues is illustrated in the sketch
proof of proposition 3 on page 6. We will reemphasise these points in the next revision.

Regarding the numbered list of questions: (1) *Exploration deterministic or stochatic?*: Stochastic—we use posterior
sampling (please see line 127 and Algorithm 1); (2) *Disconnect between pseudocode and text*: We have not found
any. Can you please provide more detail?; (3) *"[F]ailure of BDQN and UBE is surprising . . . I do not see how they
fail so much"*: Failure of UBE is predicted in Proposition 1 (please see line 100). For BDQN, we hypothesise that
before finding the reward signal, $P_Q$ depends more on the random initialisation of the NN then on the actual MDP.
Please note that the code and instructions to reproduce all experiments can be found in the supplementary material;
(4) *Section 5.3 is unnecessary*: Section 5.3 shows we reliably outperform BootDQN+Prior (the strongest competitor
of SU) on the benchmark proposed in the BootDQN+Prior paper (Osband et al., 2018)—we believe this to be one
of the strongest empirical results in our paper; (5) *"y-axis in Figure 4—clipped or is-between"*: clipped (please see
caption of Figure 4); (6) *Explanation of assumptions about state-action embeddings:* Most assumptions follow from
the definition of successor features (Dayan, 1993; Barreto et al., 2017). For the rest, please see lines 133 (unit norm),
and 142–143 (non-negativity). (8) *Are "tied actions" equivalent to "stochastic transtions"?*: No, tied actions means
that $a_1$ is always mapped to UP and $a_2$ to DOWN *in each state*, as opposed to this mapping being *different between
states*. This mapping is randomised at the beginning but then kept fix, leading to deterministic transitions (please see
lines 359-360); (9) *Are action just another input to the network:* No, the state is fed in and all actions are considered to
determine highest Q value (please see Section 4.1, and lines 592–605 in the appendix for more detail).

**R2:** We thank R2 for recognising the empirical strength of SU and the promising nature of our work in regard to future
research. R2 observes that BootDQN does not satisfy the definition of *Randomised Policy Iteration* (RPI): This is
intentional as BootDQN does not suffer from the issues discussed in Section 3 (the reasons to prefer SU over BootDQN
are so far purely practical: a significantly lower computational cost and better empirical performance). It may be
more natural to define an RPI method as an algorithm iterating *policy improvement* and *value prediction* steps while
maintaining a distribution over the values and/or policies. We will distinguish between this more general definition and
the "single policy" RPI methods in the next revision; please note that this will not affect our theoretical claims.

R2's Proposition 1 comments: (i) It is *not* the case that "the analysis fixes the policy" which, as R2 points out, would be
quite limiting. The result holds for any algorithm which employs a factorised Q function distribution with symmetric
marginals. The confusion perhaps comes from the $\pi$ superscript used in statement of Proposition 1; we will adjust
the notation in the next revision. (ii) It indeed may seem that function approximation will lead to high correlation
between Q values of nearby states. However, our experiments in Section 5.1 show that BDQN, which uses neural
network function approximation, fails to outperform the uniform exploration policy (this phenomenon was present in all
architectures we tested). As mentioned in response to R3's question (3), we hypothesise this is due to $P_Q$'s dependence
on initialisation before finding the reward signal. SU can be seen as a simple fix which can leverage information about
transitions even without observing any rewards. We agree that gaining thorough theoretical understanding of why
BDQN fails in Section 5.1 is an interesting direction of future research.

R2's Proposition 2 comments: The purpose of this proposition is to prove that "propagation of uncertainty" is not
*necessary* to satisfy our Definition 2. That propagation of uncertainty is not *sufficient* for effective exploration is shown
by Proposition 1 and experimentally in Section 5.1, meaning that SU's and BootDQN's success cannot be ascribed
to propagation of uncertainty when posterior sampling is used. However, we do agree with R2 that matching PSRL's
distribution over policies directly would be preferable to satisfying Definition 2. Doing so in a computationally tractable
way in large scale settings remains a challenge though which is why all contemporary algorithms (including SU) employ
approximations. We will clarify these points in the next version of our manuscript.

**R1**: We thank R1 for recognising SU's strong empirical performance, and our contributions to the ongoing theoretical
exploration of PSRL. We are glad R1 highlighted relative simplicity of SU which may lead to its wider adoption, and
are thankful for the suggestions regarding writing which we will implement in the next revision of our paper.

[Meta-Review · NeurIPS 2019]

From the discussion, the reviewers appreciated the precisions made in the rebuttal. They have indicated what they would like to see improved in a revised version, in particular a clearer presentation.